# Budding yeast complete DNA synthesis after chromosome segregation begins

Tsvetomira Ivanova [1,2,8], Michael Maier[1,2,8], Alsu Missarova[2,8], Céline Ziegler-Birling[3], Monica Dam[3], Mercè Gomar-Alba[3], Lucas B. Carey [2,4✉] & Manuel Mendoza [1,2,3,5,6,7✉]

To faithfully transmit genetic information, cells must replicate their entire genome before division. This is thought to be ensured by the temporal separation of replication and chromosome segregation. Here we show that in 20–40% of unperturbed yeast cells, DNA synthesis continues during anaphase, late in mitosis. High cyclin-Cdk activity inhibits DNA synthesis in metaphase, and the decrease in cyclin-Cdk activity during mitotic exit allows DNA synthesis to finish at subtelomeric and some difficult-to-replicate regions. DNA synthesis during late mitosis correlates with elevated mutation rates at subtelomeric regions, including copy number variation. Thus, yeast cells temporally overlap DNA synthesis and chromosome segregation during normal growth, possibly allowing cells to maximize population-level growth rate while simultaneously exploring greater genetic space.

[1] Centre for Genomic Regulation (CRG), The Barcelona Institute of Science and Technology, Barcelona, Spain. [2] Universitat Pompeu Fabra (UPF), Barcelona, Spain. [3] Institut de Génétique et de Biologie Moléculaire et Cellulaire, Illkirch, France. [4] Center for Quantitative Biology and Peking-Tsinghua Center for the Life Sciences, Academy for Advanced Interdisciplinary Studies, Peking University, Beijing, China. [5] Centre National de la Recherche Scientifique, UMR7104 Illkirch, France. [6] Institut National de la Santé et de la Recherche Médicale, U964 Illkirch, France. [7] Université de Strasbourg, Strasbourg, France. [8] These authors contributed equally: Tsvetomira Ivanova, Michael Maier, Alsu Missarova. ✉email: lucas.carey@pku.edu.cn; mendozam@igbmc.fr

Eukaryotic cells must complete DNA replication before chromosome segregation in order to maintain genomic stability. Complete replication is thought to be ensured by the temporal separation of DNA synthesis (S-phase) from mitosis (M-phase)[1]. The ordering of S and M phases is established by increasing levels of cyclin-dependent kinase (Cdk) activity during the cell cycle[2] and is enforced by checkpoints that inhibit chromosome segregation when cells are exposed to severe replication stress[3] or when bulk DNA replication is delayed[4]. However, it is unclear how cells could detect unreplicated DNA during unperturbed conditions, and what the detection thresholds of such mechanism may be. Interestingly, cancer cells exposed to mild DNA replication stress perform DNA synthesis in early mitosis and possibly even in the subsequent G1[5–7], raising the possibility that DNA synthesis and mitosis may overlap during normal cell cycles. Supporting this view, certain budding yeast mutants can enter mitosis in the presence of unreplicated DNA[8–12]. Thus, to what extent eukaryotic cells temporally separate DNA synthesis and segregation under physiological conditions remains an open question.

Here we present evidence that 20–40% of normally dividing yeast cells enter mitosis before the completion of DNA replication. DNA synthesis is inhibited in early mitosis and resumes in anaphase, when Cdk activity drops. In addition, our data suggest that anaphase DNA synthesis of chromosome end regions may contribute to their high mutation rates and rapid evolutionary diversity.

## Results

**DNA synthesis in late mitosis promotes nuclear division.** To test if DNA synthesis occurs during mitosis in unstressed cells, we arrested yeast in metaphase via depletion of the anaphase promoting complex activator Cdc20 and measured incorporation of the nucleotide analogue 5-ethynyl-2′-deoxyuridine (EdU) as cells were, or were not, released into a G1 arrest (Fig. 1a). Cells held in metaphase showed no nuclear EdU signal after a 60-min pulse, although they showed bright spots in the cytoplasm, likely reflecting EdU incorporation into mitochondrial DNA (Fig. 1b). Consistent with this interpretation, inhibition of mitochondrial DNA synthesis with ethidium bromide[13,14] reduced cytoplasmic EdU foci in metaphase-arrested cells (Supplementary Fig. 1). In contrast, cells released from metaphase into G1 arrest (by expression of Cdc20 and exposure to alpha-factor) incorporated EdU into both mitochondria and the nucleus (Fig. 1b and Supplementary Fig. 1). In G1-arrested cells, the nucleus was divided into a DAPI-rich and a DAPI-poor region, which corresponded to the nucleolus (visualised with fluorescently labeled Net1; Fig. 1c). EdU incorporation was higher in G1 than in metaphase cells in both DAPI-rich and DAPI-poor nuclear regions (Fig. 1b and Supplementary Fig. 1). Detection of EdU was not affected by differences in cell cycle stage such as chromosome condensation (which in budding yeast, is highest in late anaphase[15]), since EdU incorporated during S phase was detected with similar efficiency in mitosis and interphase (Supplementary Fig. 2). Nuclear EdU incorporation was not associated with DNA damage owing to metaphase arrest, since we did not detect Rad53 phosphorylation in metaphase or the following G1 after prolonged depletion of Cdc20 (Supplementary Fig. 3). Moreover, freely cycling unstressed cells exposed to a shorter (10 min) EdU pulse also showed significant nuclear EdU incorporation in late mitosis and G1, whereas mitotic cells with actively segregating nuclei did not (Supplementary Fig. 4). These observations suggest that metaphase is refractive to nuclear EdU incorporation, but that some incorporation occurs between metaphase and the following G1 even in freely cycling, unstressed cells.

Nuclear EdU incorporation in G1 cells may reflect DNA synthesis in the nucleus and/or increased nuclear import of nucleotides. To test if EdU incorporation depends on DNA synthesis, we sought to inactivate DNA replication in metaphase arrested cells. Because the catalytic activity of DNA polymerase epsilon is not absolutely necessary for DNA synthesis[16,17], we chose to deplete the catalytic subunit of DNA polymerase delta using an auxin-inducible degron. However, depletion of Pol3 in metaphase-arrested cells inhibited nuclear division (Supplementary Fig. 5). This prevented us from formally demonstrating that nuclear EdU incorporation in G1 cells depends on DNA synthesis. On the other hand, this result also suggests that mitotic DNA synthesis promotes nuclear division. To investigate this directly, cells were arrested in metaphase by depletion of Cdc20 or by treatment with nocodazole, an inhibitor of microtubule polymerization. Upon release from metaphase, DNA synthesis was inhibited by treatment with the ribonucleotide reductase inhibitor hydroxyurea (HU) or by inactivating DNA replication factors with temperature-sensitive (ts) mutations, including DNA polymerase alpha (pol1–13), delta (cdc2-1; hereafter termed pol3-ts) and epsilon (pol2-12), and the GINS complex component Psf2 (psf2-ts). Nuclear and cell division were visualised with the Histone H2B (Htb2)-mCherry reporter, and with Myo1-GFP or the membrane marker GFP-CAAX, respectively. Disruption of DNA synthesis during mitosis delayed or inhibited nuclear and cell division and triggered long-lived chromatin bridges (Fig. 2a–f). This is consistent with classical reciprocal shift experiments showing that inhibition of DNA synthesis in mitosis allows division of most, but not all cells[18]. The severity of nuclear and cell division delays varied among ts mutants, with pol2 mutant cells displaying the strongest defects, whereas psf2, pol1 and pol3 displayed relatively mild delays (Fig. 2c–e). Differences between the ts mutants might be due to polymerase-specific functions or to variations in the penetrance and inactivation kinetics of different ts alleles. Disruption of mitotic DNA synthesis with HU also delayed cytokinesis in cells without tagged histones, although with reduced severity (Fig. 2f) and causes a slight delay in the segregation of the nucleolus, labeled with Net1-GFP (Supplementary Fig. 6). Interestingly, deletion of the RAD9 checkpoint gene abolished nuclear division delays and chromatin bridge formation in response to challenges in DNA synthesis during mitosis (Fig. 2b, c). This may indicate that Rad9 responds to DNA damage during early anaphase, as suggested by a previous study[19]. Alternatively, rad9Δ strains may be less sensitive to perturbation of mitotic DNA synthesis, for example due to higher dNTP levels that accelerate fork progression during the preceding S-phase[20]. Supporting this possibility, chromosome instability mutants and checkpoint-defective rad53 cells have elevated dGTP levels[20,21], although to our knowledge, whether dNTP levels are higher in rad9 mutants than in wild-type cells is not known.

The previous results indicate that DNA synthesis may continue during anaphase during normal cell division in freely cycling cells. Supporting this notion, 45% of log-phase cells entered anaphase with single-stranded DNA, which can be associated with DNA synthesis, detected as Replication Protein A (RPA) foci (Rfa2-GFP) (Fig. 3a). In addition, freely cycling cells that started anaphase in the presence of RPA foci were delayed in nuclear division relative to cells without anaphase RPA foci (Fig. 3b, c). Thus, chromosome segregation dynamics and DNA synthesis during anaphase are associated with each other. Together, these results indicate that mitotic DNA synthesis promotes timely chromosome segregation.

**Segregation of chromatin bridges requires the mitotic exit network (MEN).** Our results suggest that cells arrested in metaphase for prolonged periods of time do not undergo DNA

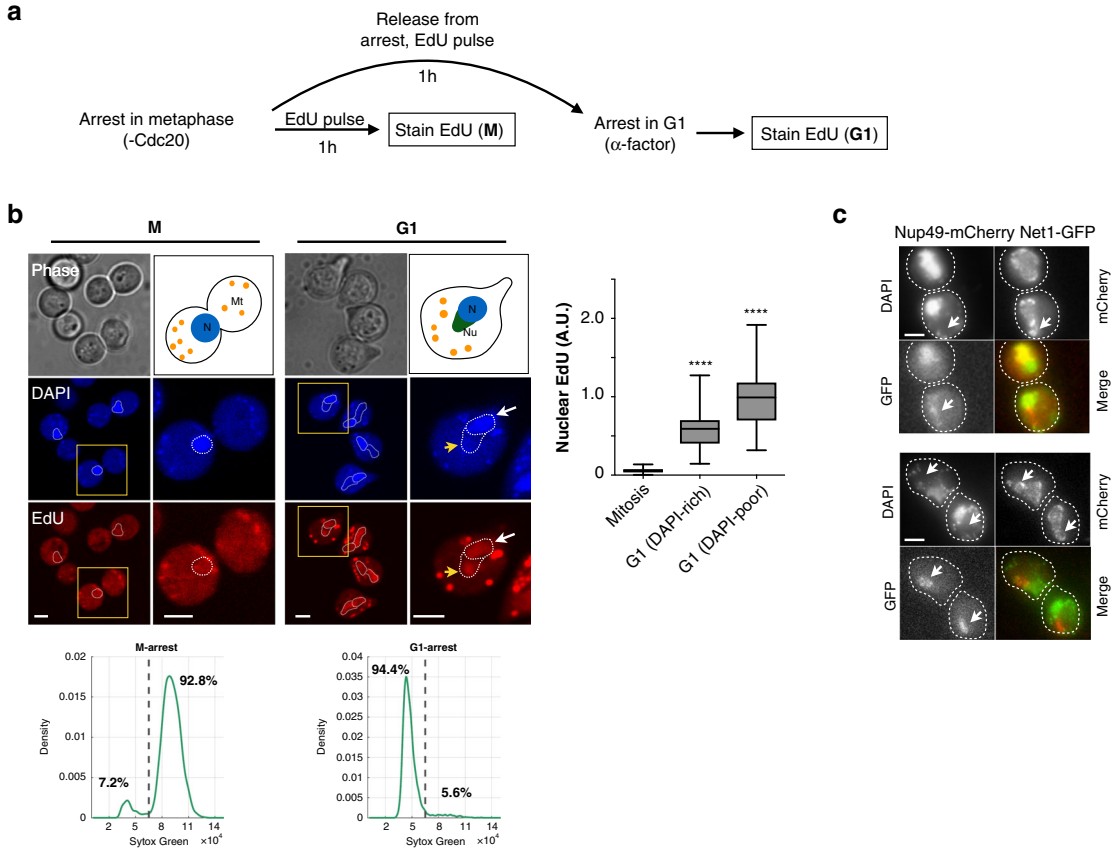

**Fig. 1 EdU nuclear incorporation occurs in G1 but not M-phase arrested cells. a** *MET3pr-CDC20* cells were grown for 3.5 h in +Met medium to block them in metaphase and treated with EdU as they were either kept in metaphase arrest or released into alpha factor-containing -Met medium for 60 min to arrest them in G1. **b** Cells blocked in G1 showed higher nuclear EdU incorporation than metaphase cells, in both DAPI-rich (white arrows) and DAPI-poor regions (yellow arrowheads). N, nucleus; Nu, nucleolus; Mt, mitochondria. ****$p < 0.0001$, two-sided $t$ test compared to metaphase. $N = 27$ cells/category. Quantification of DNA content by sytox green for cells arrested in metaphase by Cdc20 depletion or released into an alpha-factor arrest are shown in the bottom. **c** To visualise G1 nuclear morphology after metaphase arrest and release, cells expressing Net1-GFP (to label the nucleolus) and Nup49-mCherry (nuclear envelope) were arrested with nocodazole for 3 h, and released into fresh medium supplemented with alpha factor. Two independent examples are shown. Arrows point to nuclear elongations that contain Net1 but stain poorly with DAPI. This experiment was repeated two times independently with similar results. Scale bars: 2 µm.

synthesis during the arrest (Fig. 1a, b). We hypothesized that high Cdk activity inhibits DNA synthesis in metaphase, and that the inhibition of Cdk during mitotic exit enables synthesis to complete. To test this, we examined cells defective in the mitotic exit network (MEN), a kinase cascade pathway required to inactivate Cdk at the end of mitosis. Temperature-sensitive MEN mutants arrest in late anaphase at the restrictive temperature, with separated nuclei and high levels of mitotic cyclins[22]. MEN ts mutants in *TEM1*, *CDC15*, *MOB1* or *DBF2* displayed a high frequency of chromatin bridges visualized with Htb2-mCherry or the DNA dye YOYO-1, indicative of incomplete chromosome segregation in at least 40% of late anaphase cells (Fig. 4a and Supplementary Fig. 6).

Time-lapse imaging revealed defects in the segregation of telomere-proximal regions in MEN ts mutant cells. A subtelomeric region (subTel12R) was imaged via the Tet repressor fused to YFP, bound to a tet operator array inserted 20 kb away from the telomere. Spindle elongation and segregation of the rDNA repetitive locus on the same chromosome arm as subTel12R, were followed in the same cells expressing mCherry fused to the spindle pole component Spc42 and Net1, respectively (Fig. 4b). MEN inactivation in *cdc15* and *dbf2* mutants did not inhibit rDNA or subTel12R segregation, although the latter was delayed in some Cdc15-deficient cells (Fig. 4c). However, in MEN mutant cells the subTel12R sister spots got closer to each other after their initial separation, with the two sister spots collapsing into a single spot in 5% of cells (Fig. 4d). Sister spot recoiling and/or collapse occurred in cells with elongated spindles and separated rDNA masses, demonstrating non-disjunction of subtelomere regions (Fig. 4d). These results suggest that linkages between subTel12R and telomere 12R persist in MEN-deficient cells.

**Resolution of chromatin bridges requires DNA synthesis.** To determine if MEN bridges are due to persistent cohesion between replicated sister DNA molecules, we used *cdc15-as1* mutants to inactivate MEN by addition of the ATP analogue 1-NA-PP1, and a ts allele of the kleisin subunit Scc1 to inactivate cohesin. Cells were grown in the presence of 1-NA-PP1 at 25 °C to inactivate MEN, and were then shifted to 37 °C to inactivate cohesin. Bridge resolution was followed by time-lapse microscopy using Htb2-mCherry as a reporter. Chromatin bridges persisted for at least 3 h in almost 90% of *cdc15-as1* cells shifted to 37 °C in both *SCC1+* and *scc1-ts* backgrounds, indicating that MEN bridges are stable and are not due to persistent sister chromatid cohesion (Fig. 5a).

Washout of 1-NA-PP1 led to chromatin bridge disappearance during anaphase (before cytokinesis, monitored with Myo1-GFP) in the majority of *cdc15-as1* cells, demonstrating that MEN

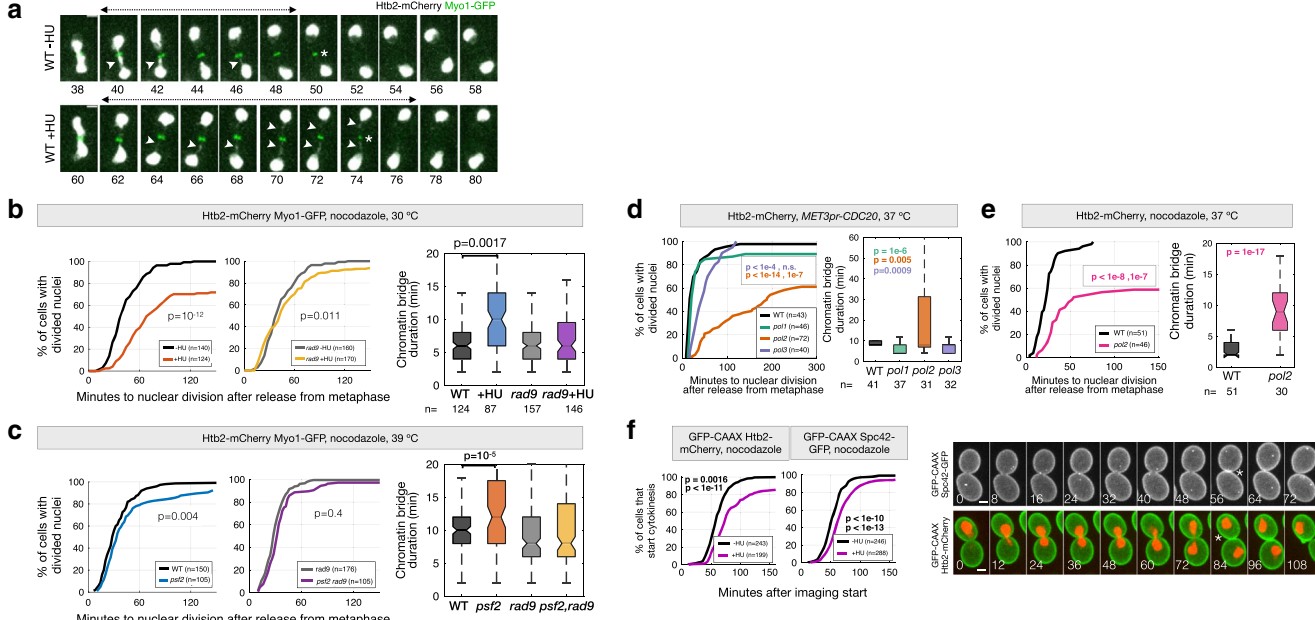

**Fig. 2 DNA synthesis in late mitosis promotes timely nuclear division. a** Cells arrested in metaphase by treatment with nocodazole and released from metaphase in fresh medium (-HU), or treated with 0.1 M HU for 30 min and released from the metaphase block in fresh medium containing HU (+HU) (see Methods). Arrowheads point to chromatin bridges labeled with Htb2-mCherry; the chromatin bridge lifetime is indicated by double-headed arrows. Asterisks mark contraction of the actomyosin ring labeled with Myo1-GFP. Images were acquired every 2 min. The time relative to imaging start is indicated in minutes. **b–e** The time of nuclear division and bridge lifetime for cells blocked in metaphase at the permissive condition, and released from the metaphase arrest after inhibition of DNA synthesis. Nuclear division is defined as the time between release from metaphase and resolution of chromatin bridges (final DNA segregation). Bridge duration is defined as the time between anaphase onset (nuclear elongation) and bridge resolution. Cells were either released from a metaphase arrest in the presence of HU at 30 °C, or arrested in metaphase at 25 °C and released at the restrictive temperature to inactivate DNA replication. Arrest conditions and fluorescent reporters are indicated at the top of each panel. **f** The time of cytokinesis (membrane closure at the bud neck, monitored with GFP-CAAX) for cells expressing the indicated reporters and released from a metaphase arrest in the presence or absence of 0.1 M HU at 30 °C. Representative cells are shown; asterisks indicate cytokinesis. The number of cells (n, pooled from at least two independent experiments) is indicated. *p* values are from two-sided Mann-Whitney, Fisher's Exact tests. Scale bars in a, f: 2 μm.

reactivation allows chromatin bridge resolution. This allowed us to test for proteins required for the resolution of chromatin bridges after MEN reactivation. Inactivation of type II topoisomerase in *cdc15-as1 top2-4* cells, by increasing the temperature to 37 °C before 1-NA-PP1 washout, did not reduce the efficiency of bridge resolution during anaphase (Fig. 5b). Thus, MEN-deficient bridges are not caused by persistent DNA catenations.

In contrast, inactivation of DNA polymerase delta or epsilon prevented bridge resolution during anaphase. Indeed, MEN reactivation in *pol3-ts* and *pol2-ts* cells caused bridge disappearance only during or after actomyosin ring contraction. This suggests that in the absence of DNA polymerase function, bridges are damaged by cytokinesis (cut phenotype), and that MEN-deficient bridges require DNA synthesis during anaphase for their timely resolution (Fig. 5c). Anaphase resolution of chromatin bridges in *cdc15-as1* cells after 1-NA-PP1 washout was not impaired by loss of the homologous recombination factor Rad51 (*cdc15-as1 rad51Δ*). In contrast, bridge resolution efficiency was reduced in cells lacking the DNA polymerase delta subunit Pol32 (*cdc15-as1 pol32*) (Fig. 5c and Supplementary Fig. 8). Consistent with DNA synthesis along chromatin bridges in MEN-deficient cells, RPA foci indicative of ssDNA are present in most *cdc15-as1* cells treated with 1-NA-PP1 during anaphase (Fig. 5d). In summary, complete chromosome segregation is promoted by MEN function and DNA synthesis during anaphase.

**Subtelomeric sequences are replicated after metaphase.** DNA synthesis during late mitosis may reflect mitotic repair of already

replicated DNA, mitotic DNA synthesis of diverse genomic regions, or mitotic DNA synthesis of specific genomic regions. To distinguish between these possibilities, we used Illumina sequencing to estimate DNA copy number[23] in cells arrested in (i) G1, (ii) metaphase, via depletion of Cdc20, or (iii) late anaphase/telophase, via inactivation of MEN (*dbf2-2*). To obtain >95% synchrony we isolated mitotically arrested cells by sucrose gradient centrifugation and used the fraction with the highest synchrony for DNA extraction and sequencing (Supplementary Fig. 9[24], Supplementary Data 1 and Methods).

To convert estimated DNA copy number into units that are more biologically meaningful and values that can be compared to those from our microscopy experiments, we used the DNA copy number ratio (CNR) between G1 and mitosis to calculate the percentage of cells in which each region of the genome is under-represented during mitosis relative to G1 (see Methods). We excluded long repeats (rDNA and telomeres) from our analysis, as in these cases DNA copy number cannot be distinguished from repeat copy number. Nonetheless, we found that chromosome ends were under-represented in a high percentage of mitotic cells: on average, 60 kb at the end of each chromosome are under-represented in metaphase (Fig. 6a and Supplementary Figs. 10 and 11a). Over 40% of cells have lower metaphase copy number in the 1 kb closest to each telomere (Supplementary Fig. 11b). Certain regions distant from telomeres were also under-represented (Fig. 6c–e and Supplementary Data 2). Most regions under-represented in mitosis correspond to a subset of late-replicating regions, most, but not all, of which are subtelomeric (Fig. 6a–d and Supplementary Fig. 12). The relationship between

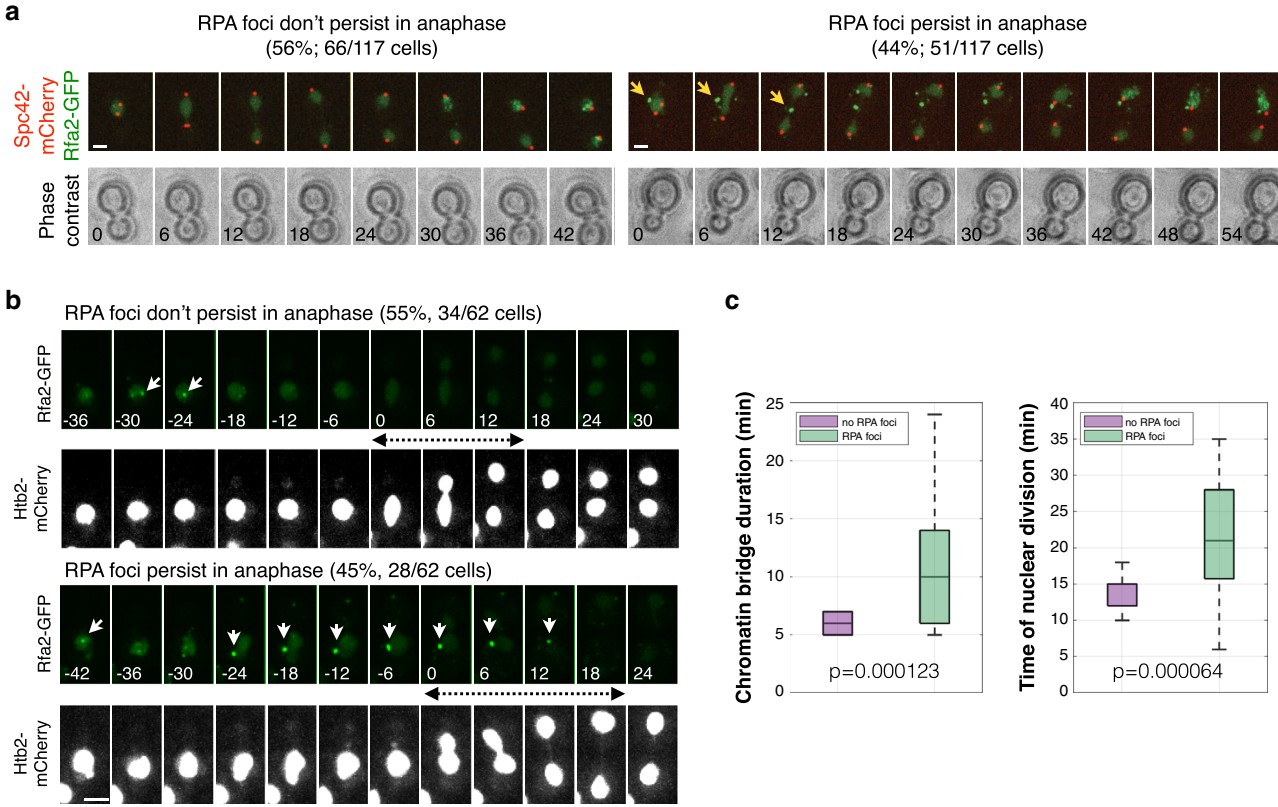

**Fig. 3 Freely-cycling cells containing single-stranded DNA during mitosis take more time to resolve chromatin bridges and divide. a** Cells expressing Spc42-mCherry to visualise the spindle poles, and Rfa2-GFP to visualise RPA foci, were grown to mid-log phase and imaged every 6 min for 3 h at 30 °C. RPA foci are present in all cells in S-phase and persist during anaphase in 44% of cells (arrows). Time from anaphase onset is indicated in minutes. Scale bar, 2 μm. **b,c** Cells with RPA foci in anaphase take more time to resolve chromatin bridges and accomplish cell division. Cells as in (**a**) but expressing Htb2-mCherry to visualise chromosomes, imaged every 5–7 min. Chromatin bridge lifetime and nuclear division were quantified for cells with and without RPA anaphase foci and represented as boxplots. On each box, the central mark indicates the median, and the bottom and top edges of the box indicate the 25th and 75th percentiles, respectively. The whiskers extend to the most extreme data points not considered outliers. Bridge lifetime was defined as the time during which the two divided nuclear masses are joined by a thinner connection (visualised with Htb2-mCherry). The timing of nuclear division is defined as the interval between the onset of nuclear elongation and bridge disappearance. $n = 62$ cells pooled from 4 independent experiments; $p$ values are indicated (two-sided Student $t$ test). Scale bars: 2 μm.

late replication timing and under-representation in mitosis is restricted to subtelomeric regions, with the exception of one 10-kb region in the middle of chromosome IV (from 981 to 989 kb) containing two transposable elements and two tRNA genes, which has a very late replication timing (>50 min) (Fig. 6d). Additionally, difficult-to-replicate regions such as transposable elements, fragile sites[25] and loci predicted to contain G-quadruplexes show significant under-representation in metaphase (Fig. 6e). We conclude that most regions near telomeres, and a relatively smaller number of loci in non-subtelomeric regions, are under-represented in mitosis and refer to these regions as "under-replicated in mitosis". Whereas G-quadruplex regions are located mainly near telomeres and therefore late-replicating, the majority of transposable elements and fragile sites are in fact telomere-distal and not late-replicating regions (Fig. 6f). Under-replication was higher in metaphase (*CDC20* depletion) than in MEN-deficient late anaphase (Dbf2 inactivation) (Fig. 6a). Under-replication for transposable elements and fragile sites becomes negligible in Dbf2-deficient cells (Supplementary Fig. 13), suggesting that these regions, which are mostly telomere-distal, complete replication after the *CDC20* execution point but before the *DBF2* execution point, i.e. in early anaphase.

**Completion of replication requires a drop in CDK activity**. To directly test if high mitotic Cdk activity inhibits DNA synthesis in

difficult-to-replicate genomic regions we determined DNA copy number in Cdc20-depleted cells after inactivation of Cdk using the *cdc28-as* allele. Inactivation of Cdk enabled Cdc20-depleted cells to finish replication of chromosome ends (Fig. 6g). We next investigated if Cdk inhibition would be sufficient to allow chromatin bridge resolution in MEN-deficient cells. To test this, we used the ts mutant *cdc28-4* to inactivate Cdk in *cdc15-as1* mutants treated with 1-NA-PP1 as in Fig. 5a. Whereas chromatin bridges resolved in only 20% of *cdc15-as1* cells shifted to 37 °C, bridge resolution occurred in ~60% of *cdc15-as1 cdc28-4* cells under the same conditions (Fig. 6h). Thus, Cdk inactivation is required to complete replication of chromosome ends during mitotic exit, preventing the formation of stable chromatin bridges.

**DNA synthesis after metaphase and high mutation rates**. There is correlation between replication timing and mutation rates in yeast and human cells[26,27] and a strong relationship between distance to the telomere and mutation rate in budding yeast[28]. Deletion of genes that are replicated in late mitosis has no apparent fitness cost, consistent with these genes being dispensable under non-challenging conditions (Fig. 7a and Supplementary Fig. 14). Genes in subtelomeric regions replicated during mitosis exhibit higher levels of intraspecies genetic diversity at the single-nucleotide level and for small insertions and deletions (<50 nt) (Supplementary Figs. 14 and 15)[29]. Further, gene

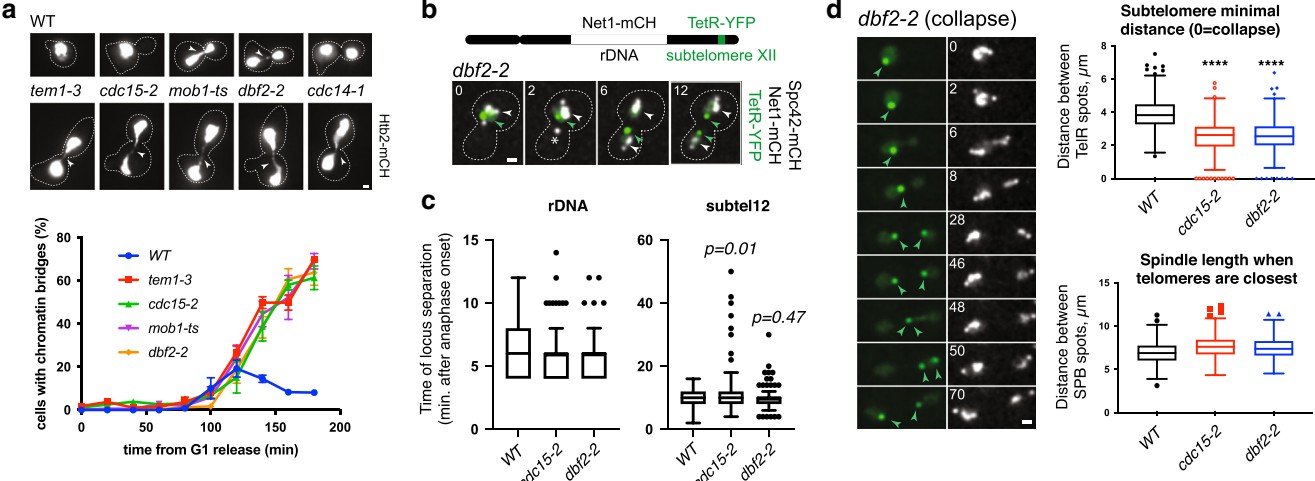

**Fig. 4 Segregation of chromatin bridges in anaphase requires the mitotic exit network. a** Quantification of chromatin bridges in MEN mutants. Cells of the indicated strains expressing Htb2-mCherry were arrested in G1 at 25 °C, released from the block at 37 °C, fixed at the indicated times, and imaged (see Methods). Images depict wt cells during the time course and MEN mutant cells 2–3 h after release from the G1 block. Arrows mark anaphase bridges. At least 100 cells were imaged per time point. The graph shows the mean and SEM of 3 independent experiments. **b** A 5-kb tetO array was inserted in subTel12R, Chromosome 12 at 1061 kb from the left telomere; the fluorescent spot is centred 20 kb away from TEL12R (green arrowheads). The rDNA locus in the same arm is labeled with Net1-mCherry (white arrowheads). The SPBs are marked with Spc42-mCherry (asterisks). **c** The time of rDNA and subTel12R separation relative to anaphase in WT and MEN mutants. $n = 202$ cells (WT); 252 (cdc15-2), 235 (dbf2-2). p values (two-sided Student T test) are indicated. **d** In MEN cells, the subTel12R sister spots get closer to each other after their initial separation. Left: an example of sister spot collapse is shown. Note that as the sister spots collapse, the spindle remains elongated and the rDNA masses separated, demonstrating non-disjunction of subtelomere regions. Top right: the minimal distance between subTel12R spots after their initial maximal separation for all cells imaged; 0 indicates collapse. Bottom right: the distance between SPBs in the same frame in which the subTel12R spots reach their minimal distance. Thus, the minimal distance between subTel12R spots is not associated with a shortening of the spindle. ****$p < 0.0001$, two-sided Student T test. Scale bars: 1 μm.

ontology analysis shows that genes under-replicated in mitosis are mostly subtelomeric, are enriched in carbon signalling and transport functions (Fig. 7b, Supplementary Data 4). This suggests that DNA synthesis during late mitosis may result in increased mutation rates of specific classes of genes.

**Delaying cytokinesis decreases subtelomeric mutation rate.** Higher mutation rates in subtelomeric late-replicating regions may occur if anaphase replication is error-prone and/or if chromatin bridges are damaged during cytokinesis. The finding that Pol32 is involved in anaphase bridge resolution (Fig. 5c) supports the first possibility, since Pol32-dependent break induced replication (BIR) is error-prone[30]. To test the second hypothesis, the counter-selectable URA3 marker was inserted at various distances from the telomere in the right arm of chromosome XIV: in two late-replicating subtelomeric loci with high under-replication in metaphase, and in two telomere-distal loci with relatively earlier replication timing that are fully replicated in metaphase. The frequency of URA3 loss (FOA-resistance) at these four loci was determined using a fluctuation test. We find that as expected, marker loss frequency increases with proximity to the telomere and further increases when DNA replication origin firing is challenged in sic1Δ strains (Fig. 7c). Loss of URA3 expression can be due to telomeric silencing and/or gene mutation. To test if cytokinesis promotes loss of URA3 expression through chromatin bridge damage, we determined the frequency of FOA-resistant colonies in cells lacking Cyk3, a cytokinesis ring-associated protein with no known roles in telomeric silencing[31]. Loss of Cyk3 delays abscission, the last step of cytokinesis, and prevents damage of chromatin bridges caused by DNA replication stress[32]. Interestingly, the cyk3Δ mutant displayed reduced URA3 loss specifically in telomere-proximal loci, but not in telomere-distal positions (Fig. 7c). These results support the hypothesis that the

increased frequency of URA3 loss at subtelomeric chromosome regions is due, at least in part, to cytokinesis-dependent DNA damage. In summary, DNA synthesis after metaphase and late anaphase segregation may contribute to high rates of evolutionary divergence in subtelomeric regions.

## Discussion

Together, our data indicates that a substantial fraction of wild-type unstressed cells complete DNA synthesis late in mitosis, long after the initiation of chromosome segregation. In particular, EdU incorporation and time-lapse imaging experiments indicate that DNA synthesis stops or slows down during metaphase and resumes during late anaphase or early G1. In addition, DNA copy number analysis suggests that difficult-to-replicate sequences are frequently under-replicated in metaphase, especially G-quadruplexes near telomeres, and a subset of transposable elements and fragile sites in non-subtelomeric regions. It will be interesting to confirm this observation with independent methods such as sequencing of DNA synthesized in anaphase / G1. We conclude that sister chromatids are still connected by unreplicated regions until late mitosis. This may explain why DNA damage in anaphase-arrested cells leads to an apparent reversion of chromosome segregation[33]. We speculate that replication forks likely stall at difficult-to-replicate sequences and that their replication is inhibited by high Cdk levels in metaphase (Fig. 8). Although high Cdk levels are compatible with bulk DNA replication in fission yeast[2], chromatin replication is inhibited in Xenopus mitotic extracts owing to Cdk-dependent replisome disassembly[34,35]. Perhaps high CDK levels block replication during budding yeast mitosis through a similar mechanism, pausing replication during metaphase to prevent damage and chromosome pulverisation[36,37]. The anaphase state may also promote completion of replication through alternative mechanisms. For instance, the Cdc14

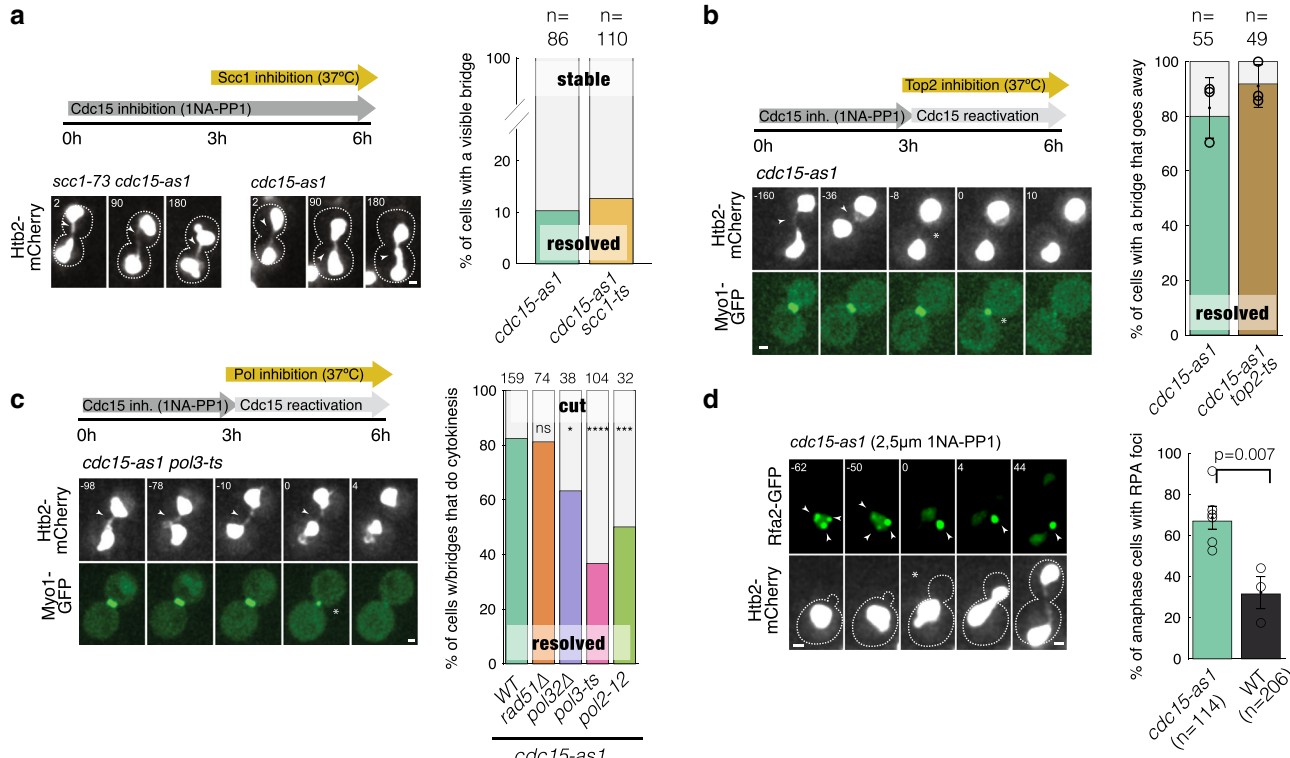

**Fig. 5 DNA synthesis is required for resolution of chromatin bridges in MEN-deficient cells. a** MEN bridges are stable even in the absence of cohesin. 1NA-PP1 was added to mid-log phase cultures at 25 °C to inactivate Cdc15. After 3 h, cultures were shifted to 37 °C, imaged by time-lapse fluorescence microscopy, and the number of cells with stable vs. resolved bridges during the next 3 h was determined. $t = 0$ corresponds to the start of imaging (temperature shift). Cells were pooled from three independent experiments. **b** MEN bridges do not require topoisomerase II for their resolution. Cells were treated as in (a) except that after 3 h, NAAP1 was removed to allow Cdc15 reactivation. Numbers indicate time (min) relative to cytokinesis. The fraction of cells resolving their bridges before actomyosin ring contraction during 3 h following washout of 1-NA-PP1 was determined. Cells were pooled from three (*cdc15-as1 top2-4*) or six independent experiments (*cdc15-as1*). **c** MEN bridges require DNA polymerases for their resolution. Cells of the indicated strains were treated and analysed as in (b). Arrowheads in a-c point to chromatin bridges and asterisks mark actomyosin ring contraction. Two-sided Fisher's exact tests: *$p < 0.05$; **$p < 0.01$; ***$p < 0.001$; ****$p < 0.0001$. **d** RPA foci persist into anaphase in MEN mutants. 1-NA-PP1 was added to mid-log phase cells at 25 °C. Representative cells are shown. Arrowheads point to Rfa2-GFP foci. The graph shows the fraction of cells with RPA foci during the first 20 min after anaphase entry. Cells were pooled from three independent experiments (two-sided *T* test across replicates $p = 0.0068$, two-sided Fisher's exact test of pooled cell counts, $p = 1.4e-09$). **a–d** scale bar: 1 μm. $n =$ number of cells.

phosphatase inhibits transcription during yeast anaphase, and this has been proposed to facilitate chromosome condensation[38]. As transcription can block replication fork progression, inhibition of transcription may not only promote chromosome condensation but also facilitate completion of replication during anaphase. In any event, our data indicate that regions under-replicated in metaphase have a last opportunity to complete their replication when Cdk levels fall below a critical threshold during mitotic exit (Fig. 8).

Why the level of mitotic under-replication varies between chromosomes, and between metaphase and anaphase, remain open questions. Intriguingly, the length of under-replication and chromosome arm length are negatively correlated ($r = -0.65$) (Supplementary Fig. 16a). One possible explanation for this correlation is that longer chromosome arms have more potential origins of replication, thus decreasing their probability of under-replication. Indeed, arms with longer under-replicated regions have fewer origins (Supplementary Fig. 16b). However, within subtelomeric regions (75 kb) both the absolute number and the density of origins (number of origins divided by the length of the chromosome arm) are positively correlated with length of under-replication; i.e. arms with more under-replicated DNA have more origins in the under-replicated region (Supplementary Fig. 16c, d).

Moreover, arms with the highest under-replication in metaphase relative to telophase tend to be shorter and have a higher ARS density (Supplementary Fig. 16e–h). That being said, firing of subtelomeric origins is highly context-dependent and variable between cells, and it is likely that not all origins fire in all cells. Therefore, the precise causal relation between under-replication, arm length, and origin density, if any, remains to be discovered.

Why do yeast not complete replication prior to mitosis? One possibility is that starting mitosis before completion of DNA replication may allow for an increased cell division rate which could outweigh occasional mutations, aneuploidy and gene loss. For instance, a decrease in the number of viable cells can be easily compensated for by an accelerated division rate (Supplementary Fig. 17). We note that this is consistent with slower proliferation causing an increase in genome stability[39]. In addition, our data raise the possibility that subtelomeric regions that replicate during mitosis, devoid of essential genes, serve as a genetic playground in which high rates of mutation and copy-number variation[40] allow cells to more rapidly explore genotypic space. We envisage two non-mutually exclusive mechanisms to account for increased mutation rates in subtelomeric regions. On one hand, mitotic DNA synthesis in these genomic regions may be intrinsically error-prone. Pol32 is required for DNA synthesis

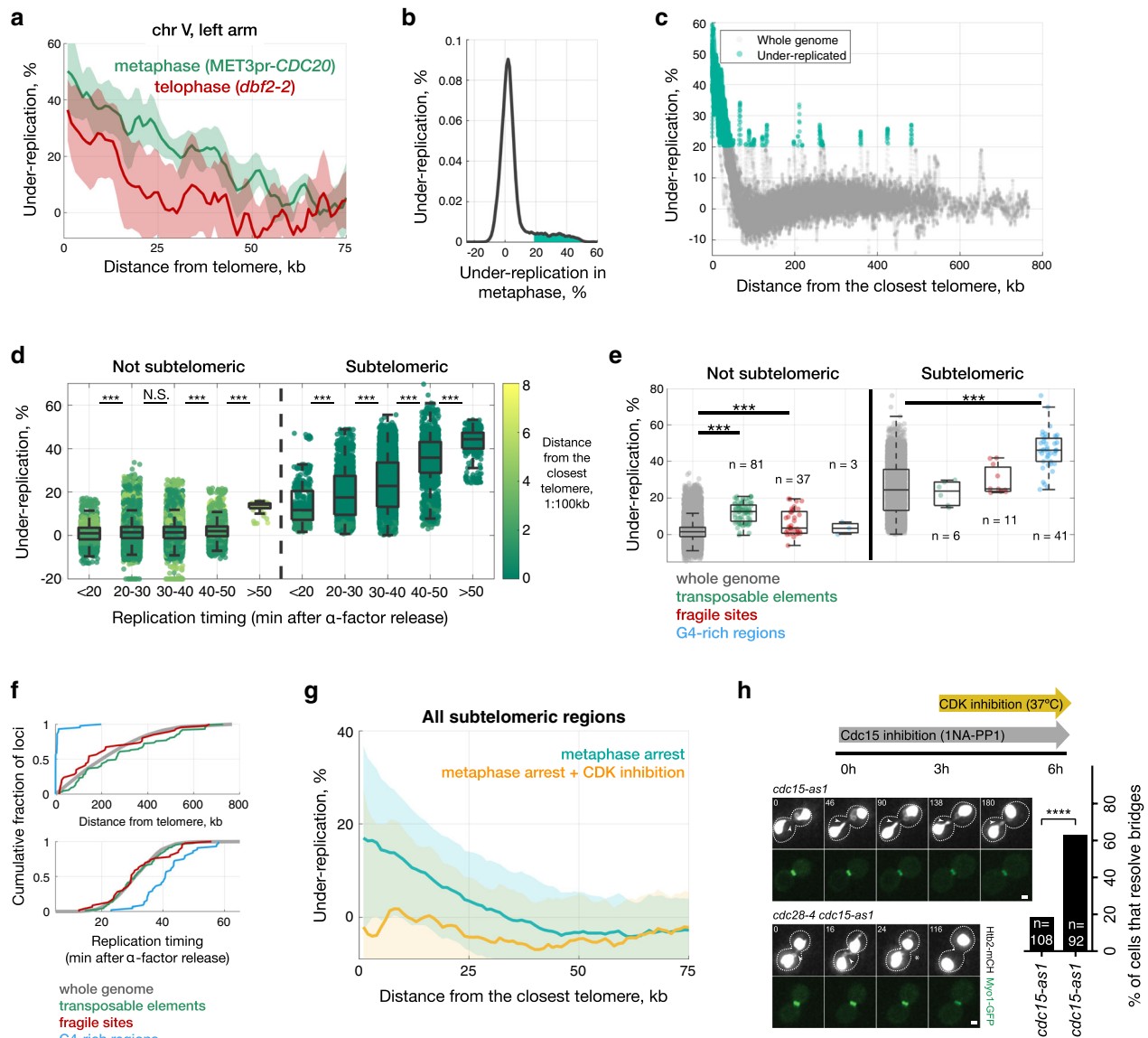

**Fig. 6 Completion of replication of subtelomeric regions and difficult-to-replicate sequences requires the drop in Cdk activity that occurs during mitotic exit. a** Under-representation of subtelomeric regions in chromosome V for metaphase (MET3pr-CDC20, in green) and telophase (dbf2-2, in red) arrests. Shadows correspond to standard deviation across biological replicates (6 for MET3pr-CDC20, 5 for dbf2-2). **b** Distribution of sequences under-represented in metaphase across the whole genome, with values greater than a threshold of 20.5% (see Methods) shaded in green (1.2 Mb, about 10% of the genome). **c** Under-representation values for all 200 bp windows throughout the genome, with significantly underrepresented genomic regions colored green. **d** Late-replicating regions show higher under-representation in both non-subtelomeric and subtelomeric regions (N.S. p > 0.05; ***p < 0.0001, Wilcoxon rank test). All 200-bp windows of measured under-replication split into bins based on their replication timing (data from[51]). Colour-code corresponds to the proximity to telomeres (1:100 kb). ~0.1% of 200 bp regions have under-representation less than −20%; for visualization we plot them at -20%. **e** Regions with high frequency of G-quadruplexes, transposable elements and fragile sites exhibit higher under-replication in metaphase (***p < 0.0001, two-sided Wilcoxon rank test). **f** Transposable elements and fragile sites are uniformly distributed across the genome and show the same replication timing as a bulk genome, whereas G4-rich regions are mainly located in subtelomeric regions and replicated later than the majority of the genome. Each transposable element and fragile site correspond to a single functional element, and G4-regions correspond to a single 200 bp window with elevated fraction of G4-sequences. **g** Genome sequencing was performed in metaphase-arrested (Cdc20-depleted) cells before and after inhibition of Cdk using the ATP analogue-sensitive mutant cdc28-as1 (see Methods). **h** Inactivation of Cdk function allows chromatin bridge resolution in MEN-deficient (cdc15-as + 1-NA-PP1) cells. Arrowheads mark chromatin bridges and asterisks bridge resolution. Time 0 corresponds to the start of imaging (temperature shift). n = number of cells is indicated. Cells were pooled from three independent experiments. ****p < 0.0001, two-sided Fisher's exact test. Scale bar: 1 μm.

initiated from DNA double strand breaks or collapsed replication forks, a process known as break-induced replication (BIR)[41], which is error-prone compared to S-phase replication[30]. Interestingly, Pol32 also promotes resolution of DNA bridges during anaphase (Fig. 5c). Thus, anaphase BIR may contribute to high mutation rates in late-replicating regions. However, Pol32-independent bridge resolution mechanisms must also exist, since a relatively higher fraction of pol32Δ cells was competent in

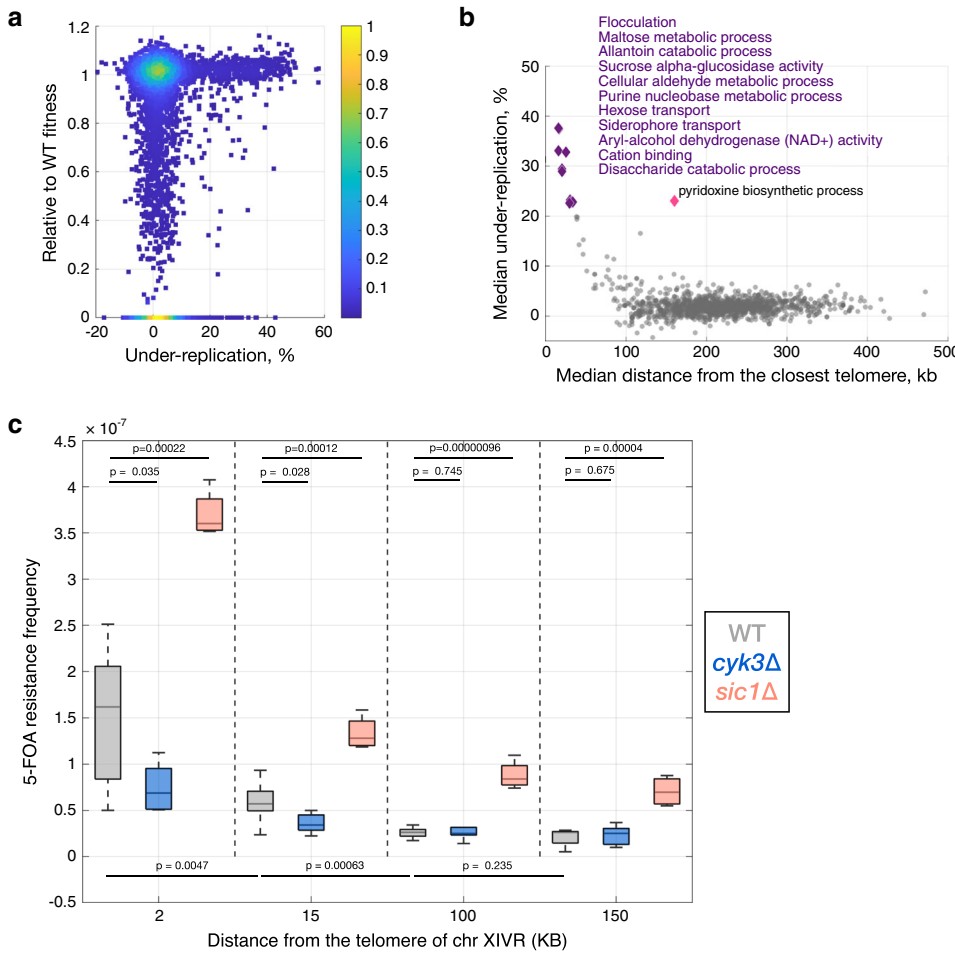

**Fig. 7 Delaying cytokinesis decreases subtelomeric mutation rate associated with under-replication in mitosis. a** Majority of genes replicated after metaphase are not essential and do not contribute to the fitness cost once knocked out. Each point represents one gene, showing the % under-replication in metaphase-arrested cells and the fitness of the deletion mutant (0 for essential genes). **b** The median under-replication for all genes in each gene-ontology (GO) term is inversely proportional to the median distance to the closest telomere. Analysis in (**a**, **b**) was performed using data from metaphase arrested *MET3pr-CDC20* cells. **c** The mutation rate is higher closer to the telomere, and a delay in cytokinesis decreases mutation rate specifically in the subtelomere. The *URA3* reporter was integrated at various distances from *TEL14R* in the indicated strains and the frequency of FOA-resistant colonies in each of the twelve strains was determined by a fluctuation test (p values from two-sided Student *T* test). Estimated under-replication values (%) and replication times (min) for the indicated loci: 2 kb, 40.6%, 36.5 min; 15 kb, 36.3%, 29.3 min; 100 kb, 0.7%, 26 min; 150 kb, -3.2%, 23.7 min. See Methods for details.

bridge resolution compared to much more severe defects after inactivation of DNA polymerase delta or epsilon. On the other hand, our finding that inhibition of cytokinesis reduces the mutation rate close to telomeres (Fig. 7c) suggests that mutations in these regions may also be caused by cytokinesis-dependent damage[32]. In summary, our findings suggest two possible causes for high mutation rates in subtelomeric regions: error-prone replication and damage of DNA bridges by cytokinesis. The first one depends on DNA synthesis after metaphase (some of which requires Pol32), whereas the second could depend on either late replication or simply on their proximity to telomeres, because telomeres segregate last, whether replicated or not. It is also possible that under-replication of subtelomeric regions could contribute to their late segregation. The precise contribution to high mutation rates from replication during mitosis and distance to telomeres remains to be established.

The idea that yeast cells complete DNA replication in anaphase has interesting parallels with mitotic DNA synthesis (MiDAS) observed in mammalian cells exposed to replication stress[6,42]. Like yeast anaphase replication, MiDAS depends on the Pol32 homologue PolD3 and does not require Rad51, opening the possibility that a form of BIR plays a role in both processes. In addition, MiDAS is inhibited in nocodazole-arrested cells (in which M-Cdk levels are highest) whereas it has been observed in prophase, when M-Cdk levels are relatively low[6]. It is not known whether MiDAS-like processes occur in anaphase in unstressed mammalian cells; however, continued DNA synthesis during late mitosis could contribute to the high frequency of ultrafine chromatin bridges, whose defective resolution is associated with genome instability[43]. Further, mitotic DNA synthesis may help explain how animal cells with rapid divisions such as in the early embryo, with no gap phases, ensure complete DNA replication.

## Methods

**Strains and cell growth**. *Saccharomyces cerevisiae* strains are derivatives of S288c except when noted (Supplementary Data 4). Gene deletions and insertions were generated by standard one-step PCR-based Methods[44]. *cdc28-as1* strains (Cdc28-F88G) were made using plasmid pJAU01-cdc28-as1 (a kind gift from Ethel Queralt) digested with Afl2 or EcoN1 for integration at the *CDC28* locus; *URA3+* clones were selected and then *ura3-* pop-outs isolated in 5-FOA plates; final clones were confirmed by Sanger sequencing. *MET3pr-CDC20* strains were made with

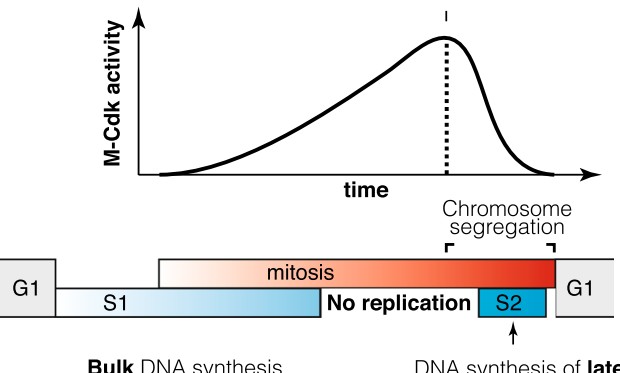

**Fig. 8 Budding yeast complete DNA synthesis after chromosome segregation begins.** The order of cell cycle phases is represented by grey (G1), blue (S1, S2) and red (M) bars. S phase is divided into pre-mitotic S1, in which the bulk of DNA replication occurs, and S2, which overlaps with chromosome segregation and in which subtelomere, transposons and fragile sites are replicated in a fraction of cells. DNA synthesis is inhibited by high M-Cdk levels during metaphase and resumes during late mitosis when Cdk levels drop and chromosome segregation is taking place.

*YiP22-MET3pr-CDC20-LEU2* (Ethel Queralt) digested with MscI for integration in the *CDC20* locus. For arrest in G1, cells were grown in YPDA (yeast extract, peptone, dextrose and adenine) medium to log phase, synchronized with 20 µg ml⁻1 alpha-factor (Sigma-Aldrich) for 2 h. For release from the G1 block, cells were washed in fresh YPDA medium at the indicated temperature. For arrest in metaphase, *MET3pr-CDC20* cells were grown in SD + Glu (synthetic defined + glucose) medium lacking methionine until mid-log phase and the media were supplemented with 2 mM methionine to repress *CDC20* for 3 h 30 min. For release, cells were washed twice and resuspended in minimal medium lacking methionine to induce *CDC20*. Alternatively, cells were incubated with 60 µg/ml nocodazole for 3 h. For Cdc15 or Cdk inhibition, *cdc15-as1* or *cdc28-as1* cells were incubated with NA-PP1 2.5 µM. For temperature shifts, cells were grown and arrested at 25 °C, shifted to 37 °C 15 min before washes, and released in prewarmed medium at 37.5 °C. Otherwise cells were arrested and released at 30 °C. To determine DNA content with Sytox-Green, we used the staining protocol from[45].

**EdU incorporation.** EdU click and DAPI staining were carried out as in[46] with minor modifications. Briefly, yeast cells (W303 RAD5 bar1 MET3pr-CDC20 GPDpr-TK(5×) ADH1pr-hENT1) expressing thymidine kinase (TK) and equilibrative nucleoside transporter (hENT1) to allow for EdU incorporation, were arrested in metaphase as indicated. Arrested cells were then pre-incubated with 25 µM 5-ethynyl-2′-deoxyuridine (EdU) and α-Factor (20 µg/ml) for 15 min. Half the culture was kept in metaphase (+Met) and the other washed and resuspended into minimal medium (–Met) supplemented with 25 µM EdU and with 20 µg/ml α-Factor to block cells in the next G1 phase. After 1 h, cells were fixed with 2% paraformaldehyde and processed for Sytox-Green staining (for flow cytometry) and for DAPI staining and EdU labeling with Alexa Fluor 647 (for microscopy)[46]. Cells were imaged in a Leica TCS SP5 confocal microscope (63× objective, 3 planes spaced 0.67 µm). Nuclear EdU incorporation was measured in maximum projections as the background-subtracted mean intensity in the nucleus (defined by DAPI staining). The background was measured in multiple small cytoplasmic areas to avoid signal from mitochondrial DNA. As not all cells are permeabilized during fixation, cells that showed no EdU signal in mitochondria were not included in the quantification (Fig. 1). Alternatively, Ethidium Bromide (30 µg/ml) was added 15 min prior to addition of EdU to reduce mitochondrial DNA; in this case all cells were included in the quantification (Supplementary Fig. S1).

EdU incorporation in freely cycling cells was carried out in the same manner as above, in cells with wild-type *CDC20* (w303, RAD5 bar1 GPDpr-TK(5×) ADH1pr-hENT1), except that cells were pre-grown in SCD to mid-log phase and α-Factor (20 µg/ml) was added for 30 min to prevent cells already in G1 from entering into S phase. Cells were then incubated in EdU (25 µM) for 10 min; EdU was visualized with Alexa 647 and nuclei were visualised with DAPI[46].

**Microscopy.** For time-lapse imaging, cells were plated in minimal synthetic medium on concanavalin A–coated (Sigma-Aldrich) Lab-Tek chambers (Thermo Fisher Scientific). Imaging was performed in a pre-equilibrated temperature-

controlled microscopy chamber, using a spinning-disk confocal microscope (Revolution XD; Andor Technology) with a Plan Apochromat 100×, 1.45 NA objective equipped with a dual-mode electron-modifying charge-coupled device camera (iXon 897 E; Andor Technology). Time-lapse series of 4.5 µm stacks spaced 0.3 µm were acquired every 2–8 min. The time interval did not affect nuclear division or chromatin bridge lifetimes. Images were analyzed on 2D maximum projections. Maximum projections are shown throughout. For YOYO-1 quantification of chromatin bridges, cells were fixed in 4% formaldehyde for 30 min, washed once in PBS and re-suspended in 5 mg/ml zymolyase in P solution (1.2 M Sorbitol, 0.1 M potassium phosphate buffer pH: 6.2) for 1 min. Cells were spun down, taken up in P-Solution + 0.2% Tween 20 + 100 µg/ml RNASe A and incubated for 1 h at 37 °C. After digestion, cells were pelleted and taken up in P-Solution containing 25 µM YOYO-1. Cells were imaged on an Andor Spinning disk microscope (15 planes spaced 0.3 µm).

**Western blot analysis.** Approximately 10⁸ cells were collected, resuspended in 100 µL water and, after adding 100 µL 0.2 M NaOH, they were incubated for 5 min at room temperature. Cells were collected by centrifugation, resuspended in 50 µL sample buffer, and incubated for 5 min at 95 °C. Extracts were clarified by centrifugation, and equivalent amounts of protein were resolved in an SDS-PAGE gel and transferred onto a nitrocellulose membrane. The primary antibodies used were anti-Cdk1 (PSTAIR) (#06-923, Merck Millipore) at a dilution of 1:2000, and anti-Rad53 ab104232 (Abcam) at a dilution of 1:8000. Blots were developed with anti-rabbit IgG Horseradish Peroxidase conjugate (Thermo Fisher Scientific, Cat. No: 31460) diluted 1:20,000 using the Supersignal West Femto Maximum Sensitivity Substrate (Thermo Scientific).

**Quantification of under-replication across the genome by genome sequencing.** To identify genomic regions that are under-replicated in mitosis, we performed whole-genome Illumina sequencing of cells arrested in metaphase (MET3pr-CDC20) or anaphase (dbf2-2). Each strain was also arrested in G1 phase (alpha factor arrest). *MET3pr-CDC20* cells were grown to mid-log phase in SC-Met, arrested in G1 with alpha factor at 25 °C for 3 h, then washed and shifted to YPDA + Met for 3 h at 37 °C to arrest in metaphase. *dbf2-2* cells were grown in YPDA, arrested in G1 with alpha factor at 25 °C for 3 h, then washed into YPDA at 37 °C to arrest in anaphase. For data shown in Fig. 6a–e, cells were harvested by centrifugation for 5 min, washed twice in PBS, resuspended in water and layered on a 10%-40% sucrose gradient. After centrifugation at 400 g for 4 min, the top layers were removed, the remaining 10 ml of the gradient were split into two fractions and the pellet was taken up in 1 ml water to form the third fraction. Aliquots were removed from all three fractions to assay for cell cycle synchrony and the rest was pelleted and frozen at −80 °C. Cell purification steps were performed at 37 °C. The fraction with the highest synchrony was used for phenol-chloroform-isoamyl DNA extraction. For data shown in Fig. 6g, cells were treated as above, except that Cdc20-depleted cells were shifted to 37 °C during 15 min to inactivate *CDC12*, and then incubated with 1-NAA-PP1 (1 µM) for a further 30 min before DNA extraction with a gDNA extraction kit (Zymo Research). Cell synchronisation was assessed by flow cytometry and by microscopy (DAPI staining).

DNA sequencing was performed using Illumina HiSeq Sequencing V4 Chemistry with PCR-free library prep and paired-end 50-bp reads. Reads were trimmed using Trimmomatic default parameters (java -Xms3G -Xmx3G -jar trimmomatic.jar PE -phred33)[47]. Mapping was performed onto the R64 reference genome[48] of S. cerevisiae using Burrows-Wheeler Aligner with default parameters (bwa mem)[49]. Reads were then sorted and improperly paired reads discarded (samtools view -F 4) using Samtools[50]. We generated 200-bp non-overlapping windows across the genome (bedtools makewindows) and computed coverage for each window (bedtools coverage). Detailed and documented pipeline can be found in the Makefile on github and in the Supplementary Information.

Normalized coverage for each window was computed separately for each sequenced sample and each chromosome by dividing the read coverage for each window on that chromosome by the mode of all coverage values for all windows on that chromosome (copy number or CN). To each CN we applied a local regression using a weighted linear least squares and a first degree polynomial model (with a 5 kb span) to decrease the impact of technical noise. For each mutant we calculated copy number ratio (CNR) for each 200-bp window as normalized copy number in M-phase divided by the normalized copy number in G1-phase. The percent of cells that have not yet replicated a given region of the genome (under-replication), is defined as 2×(1-CNR).

We expect the measured copy number in G1-arrested cells to be 1 across all loci in the genome. However, multiple factors, such as GC-content and mappability, make this not true[23]. We assume that the effect of such factors will be similar between identical loci in different cell-cycle stages (G1 and M) and therefore locus-specific effects on measured read counts can be eliminated through normalization of CN in M and G1 phases[23]. This normalized copy number is the copy number ratio (CNR). To calculate the percent of cells that have not yet replicated a given locus, from CNR, we do the following. Let us denote that $X$ ($0 \leq X \leq 1$) is the fraction of cells that yet didn't replicate a given locus in M-phase. Accordingly, $1 - X$ is the fraction of cells that have finished replication of this locus. Therefore, in a mixed population of cells, some of which have finished replication, and others that haven't, the locus will have a total representation of $X + 2 \times (1 - X)$ copies.

Next, we assume that most of the genome has completed replication in all cells before entering M phase. The majority of loci (which corresponds to the mode of coverage which is used as normalization factor) will have total representation of 2 copies in M phase. All together, it gives us the linear equation: $CNR = (X + 2 \times (1 - X))/2$. Therefore, $X$, the fraction of cells that have not yet replicated a given locus, is $X = 2 \times (1 - CNR)$.

The above was performed for six biological replicates for each strain and condition (see Supplementary Table S1), one outlier replicate was removed, and the median value of under-replication across replicates was assigned as a final value for each 200-bp window. All raw FASTQ files together with smoothed CN and datasets with calculated statistics for each 200-bp genomic window are on github and NCBI GEO.

Finally, we excluded the rDNA region from the subsequent analysis due to inability of distinguishing DNA copy number from repeat copy number. Specifically, we discarded the region between 450.6 and 491 kb of chromosome XII, which encompasses two copies of RDN1 that are present in the reference genome and RDN5-5. We also discarded nine 200 bp windows that show extremely low under-replication (< 100%) due to being very close to existing deletions in our strain, which affects both the estimate for the under-replication in these regions and the estimate of the skewness of the distribution of under-replication. Finally, we discarded all regions that show a total copy number of 0 for all G1 and M replicates (around 5.6 kb in total) that are likely deleted in our strain.

### Analysis of under-replicated regions.

To define which genomic regions (at a resolution of 200-bp) were significantly under-replicated we compared the actual CNRs between M-phase and G1-phase against technical noise. To generate a technical and biological noise CNR dataset we took CN data from pairs of G1-arrested biological replicates and calculated CNR (CNR-G1-control) between them in the same manner we did for M- and G1-phase (overall 15 pairs, for each pair - 2 CNR vectors: CN(replicate #1)/CN(replicate #2) and CN(replicate #2)/CN(replicate #1)). Then for each 200-bp window, starting from the first 200-bp from each telomeric end, we compared values of actual CNR vectors (5 in total, one for each repeatable replicate) against CNR-G1-control vectors at this window (Student T-test). The first window, starting from the telomeric end, where difference was not significant (*p* value < 0.05) or the median for CN-ratios between samples in G1 was higher than median of actual CNR-value, was marked as the end of under-replicated region.

To choose a threshold value of under-replication at which regions of the genome are under-replicated in metaphase we used the under-replication distribution of all 200-bp windows. This distribution has a heavy right tail and positive skewness. The threshold value selected to classify loci as being under-replicated is the highest under-replication value where skewness of values lower than the selected value is 0.

### Quantification of under-replication for genomic features.

To extract a single value for different genomic features of varying length, we first extracted coordinates of the features of interest. We compiled a dataset from different sources, including SGD (https://www.yeastgenome.org for centromeres tRNAs, transposons and open reading frames (ORFs) in general) and OriDB (http://cerevisiae.oridb.org) for ARSs. ARSs marked as dubious were excluded from the analysis. Additionally, we added regions that were likely to undergo loss of heterozygosity (LOH) as well as duplications and deletions during cell growth[25]. We denoted as fragile sites regions marked as either interstitial or terminal deletions or duplications in[25]. For all collected features, we extracted a single value of under-replication from the 200-bp resolution CNR vector using bedtools software (bedtools coverage). Replication timing data were extracted from[51] and spline interpolation was used to define a single replication timing value for each 200-bp window. To predict the occurrence of G-quadruplexes throughout the genome, we used G4-Hunter[52] with a window size of 25nt (the default) and a score threshold of 1 (python G4Hunter.py -i yeastgenome.fasta -o S1_W25 -w 25 -s 1). For each 200 nt window we took the sum of all sequences that passed the score threshold (bedtools map -c 5 -o sum -a genome_windows_by_200.bed -b< G4HunterOutputFile.nts>). To calculate average under-replication and distance to telomeric start on gene ontology level, we extracted database with records of all ontologies and corresponding ORFs (not nested) from SGD, and for each ontology we calculated mean and CV (std/mean) for under-replication and distance to telomeric start for all genes in each GO term. The relative fitness for all essential genes was assigned as 0 while the measured fitness for non-essential genes are from[53]. To extract representative metrics of genetic divergence between strains (mutations and copy-number variation) we used whole-genome sequences of 85 strains from[54]. For each 200 bp-window, we calculated the frequency of SNPs and Indels. The frequency is the # of positions in each 200 nt window that have a SNP or Indel in at least one strain.

### Stochastic simulation of the relation between doubling time and chance of death.

Each cell in the figure shows the results from one simulation. Each simulation begins with 1000 cells, all of which have the same doubling time (80–100 timesteps (min), *y*-axis). The simulation proceeds one timestep (min) at a time, and when each cell reaches its doubling time, it divides symmetrically into two cells. Every cell has an *X*% chance of dying (*x*-axis). The simulation is run for 300 timesteps, the total number of cells at the end is measured, and the number of population doublings calculated (the color of each cell).

### Fluctuation assay.

To estimate the mutation rate across different loci and backgrounds we used the Luria-Delbrück Fluctuation Assay[55,56]. We used URA3 from *S. bayanus* as a target for negative selection. TEFpr-yeGFP-URA3 construct was integrated via homologous recombination into four loci: 2, 15, 100, and 150 kb from the telomeric end of the right arm of chromosome XIV in BY4741. Additionally, in all four strains we deleted SIC1 or CYK3 by inserting TEFpr-Hyg-TEFterm cassettes into the corresponding ORFs via homologous recombination. We decided on this order of strain construction due to the genome instability of *sic1* strains.

Once strains were generated and confirmed using colony PCR and flow cytometry for the GFP channel, we performed the fluctuation assay. Strains were pre-grown from the glycerol stock in SCD-Ura overnight, diluted 1:10,000 in SCD and split into wells in a 96 well plate (200 μl each). Plates were left for 48 h in a 30 °C room without shaking. Next, we spotted separate cultures (all 200 μl) onto SCD + 5-FOA (1 g/L) agar plates. 5-FOA is toxic for URA3 strains, but *ura3* mutants can grow. The number of plated spots varied from 35 to 42 between different strains and replicates. We kept agar plates at 30 °C for seven days to allow cells that acquired mutations in the GFP-URA3 locus (or cells with other unaccounted 5-FOA resistance events) grow into colonies. Given that the distribution of number of mutated colonies across different spots follows a Poisson distribution, the number of expected 5-FOA resistance events is $-\ln(\rho 0)$, where $\rho 0$ is the fraction of spots with 0 microcolonies after 7 days of growth. Accordingly, the final 5-FOA resistance frequency (a proxy for the mutation rate) is the number of colonies on 5-FOA plates divided by the number of cells plated. To determine the total number of cells plated, for each strain we used three wells to measure the optical density of the culture (OD). To account for potential variance in the number of cells between different cultures with the same OD, for each strain we plated cells of OD = 0.0001 (onto YPD plates) and counted the number of microcolonies (this calibration plating was performed three times for each strain). Overall, the fluctuation assay was performed at least four times for each strain.

### Software.

All data analysis was done in Matlab 2018b or Prism V8. bedtools was used for preprocessing sequencing data. iQ Live Cell Imaging software (Andor Technology) was used for image acquisition.

### Statistics and reproducibility.

Unless indicated otherwise, the central line in box plots shows the median and the bottom and top edges of the box indicate the 25th and 75th percentiles, respectively. The whiskers extend to $q3 + 1.5 \times (q3 - q1)$ and to $q1 - 1.5 \times (q3 - q1)$, where q1 and q3 are the 25th and 75th percentiles of the sample data, respectively. These are the MATLAB 2018b default settings for boxplot. Microscopy experiments were repeated at least two times, on two different days. Sequencing experiments were repeated at least six times.

For microscopy, each experiment was repeated at least twice on two different days. For sequencing, at least six biological replicates were performed for all sequencing experiments.

### Reporting summary.

Further information on research design is available in the Nature Research Reporting Summary linked to this article.

## Data availability
All strains, plasmids and data are available from the authors upon reasonable request. Raw and processed sequencing data are at NCBI GEO accession GSE117268.

## Code availability
Data and code used in this work are available at https://github.com/amissarova/Mendoza__ReplicationEvolution.

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

## Acknowledgements

We thank the UPF/CRG Flow Cytometry core, the CRG sequencing and microscopy core facilities, the IGBMC Imaging Centre, Giulia Rancati, and Wah Ing Goh and Graham Wright at the A*STAR Microscopy Platform. We thank Yves Barral, Damien Coudreuse, Bruce Futcher, Travis Stracker, Jordi Torres-Rosell, Jenny Wu and Philip Zegerman for helpful comments, Charlie Boone and Phil Hieter for sharing yeast strains, and Life Science Editors for editorial assistance. This study was supported by Ministerio de Economía y Competitividad (MINECO) (BFU2015-68351-P) and AGAUR (2014SGR0974 & 2017SGR1054) grants to L.B.C. and the Unidad de Excelencia María de Maeztu, funded by the MINECO (MDM-2014-0370); the European Research Council (ERC) Starting Grant 2010-St-20091118 to Ma.M., the Spanish Ministry of Economy and Competitiveness, 'Centro de Excelencia Severo Ochoa 2013–2017', SEV- 2012-0208 to the CRG and the grant ANR-10-LABX-0030-INRT, which is a French State fund managed by the Agence Nationale de la Recherche under the frame programme Investissements d'Avenir ANR-10-IDEX-0002-02 to the IGBMC.

## Author contributions

T.I., Mi.M., A.M., C.B.-Z., M.D. and M.G.-A. performed experiments. T.I., Mi.M., A.M., C.B.-Z., M.D., M.G.-A., L.B.C. and Ma.M. analyzed the data. L.B.C. and Ma.M. acquired funding, supervised the study and wrote the manuscript with input from all authors.

## Competing interests

The authors declare no competing interests.
