## [Peer Review File · Nature Communications]

Reviewers' comments:

Reviewer #1 (Remarks to the Author):

Ivanova and colleagues show that a small, but significant amount of DNA replication occurs out of S phase, in late stages of the budding yeast cell cycle. Authors show that cells in metaphase do not polymerize DNA, but DNA synthesis is resumed later in anaphase. This synthesis is necessary for proper nuclear division. High Cdk activity in metaphase is found to prevent completion of DNA replication while nuclear segregation requires DNA polymerases and MEN function. Deep sequencing analysis of G2/M cells identifies under-replicated regions in metaphase, which mainly comprise subtelomeric regions, transposable elements, fragile sites and G4-rich regions. Finally, an *in silico* analysis links late-replicating regions to high mutation rates, pointing to a possible evolutionary benefit for delaying replication until late mitosis.

The paper is convincing, some of the approaches are remarkable and data is technically sound. Overall, the manuscript supports the rather unexpected conclusion that Cdk activity in metaphase prevents completion of replication. This is an observation that breaks established models and should therefore merit its publication in Nature Communications. Although the manuscript does not have the mechanistic insight into how Cdk activity controls replication in mitosis, the observations are highly relevant and represent an important contribution to the fields of chromosome replication, segregation and cell cycle control. My main criticism is that the manuscript is extremely concise. From my point of view, a longer version, including appropriate experiment descriptions and discussions, would make it more accessible.

There are some minor issues, mostly related to data shown in supplementary figures or not properly described in the main text:

1. It is not clear how authors differentiate EdU incorporation in DAPI-rich and DAPI-poor areas. Text is written as if DAPI-poor corresponds to the nucleolus. Have authors co-localized high EdU incorporation with nucleolar markers?
2. Also related to DAPI-poor areas, if the nucleolus is a major site for late mitotic replication, does polymerase inhibition in mitosis preferentially affect rDNA segregation?
3. Fig 1a indicates that almost all cells increase EdU signal as they transit from metaphase to G1. Since not all chromosomes incorporate EdU, one would expect to localize EdU signal at specific sites (foci) rather than homogeneously dispersed throughout the nucleus/nucleolus.
4. One final caveat about EdU incorporation is whether EdU is really polymerized into a new DNA strand or EdU uptake into the nucleus/nucleolus is simply upregulated as cells exit mitosis. Authors could address this by inactivating polymerases, as in Fig 1C.
5. Authors should clarify at what stage in mitosis does DNA synthesis resume. Experiments shown in Fig S2 suggest it is not in anaphase (lowest level of EdU incorporation), while Fig 1 indicate it is not in metaphase. Then, it has to be after anaphase...? There is no example for "Stage 3" cells, also able to incorporate EdU. Are they in metaphase? Early anaphase?
6. Authors suggest the presence of a Rad9-dependent checkpoint, which delays nuclear division in response to replicative stress in M phase. This is nicely shown in Fig S4A, and described in just one sentence. However, there are some issues in the Fig S4B (duration of chromatin bridges) that require further clarifications. Specifically, authors should explain why bridges in *rad9* cells (no HU) take longer to dissolve than in wt cells (Fig S4B). It is also quite puzzling that treatment with HU reduces duration of chromatin bridges in *rad9* cells. In addition, the graph does not show big differences between chromatin bridge duration in wt and *rad9* mutant cells when cells are treated with HU (Fig S4B). This does not fit with a model in which *rad9* deletion advances nuclear division

in HU (Fig S4A).

7. Related to point 5 and 6, it is also unclear at what stage Rad9 arrests cells. Is it before anaphase entry? This is relevant, as a checkpoint responding to defects in DNA polymerization should become active after DNA polymerization has been attempted.

8. Fig S4C and D. It seems that polepsilon and GINS promote chromatin bridge resolution and nuclear division in mitosis, while polalpha and poldelta do not. Although not all its alleles used may affect DNA replication to the same extent, the (possible) differential contribution of DNA polymerases should be discussed in the text.

9. Figure S4 legend is incomplete and difficult to follow. Also, there are no references in the main text to some parts in Fig S4.

10. Figure 2B. There is no control for no cdc15-as reactivation. Cells are monitored for extended periods. Authors should discard that changes observed during 1NA-PP1 washout are a matter of time, not dependent on Cdc15 reactivation.

11. Fig S13. Over-expression of CLB2. There are not big differences in replication timing between cultures expressing or not CLB2. An effect of sustained high mitotic Cdk activity would be better addressed by deep sequencing or accumulation of anaphase bridges, probably more sensitive than FACS.

12. The observation that Cdk activity prevents completion of DNA replication in mitosis is unexpected, given its role in promoting DNA synthesis in S phase. The real question here is how Cdk activity blocks completion of replication. Although authors speculate about direct Cdk inhibition of DNA synthesis (fork pausing during metaphase to prevent chromosome pulverization), an indirect effect is also possible. For example, there are reports linking Cdc14-dependent Cdk downregulation to RNA polymerase transcription and chromosome segregation. As transcription can block replication fork progression, a modification of the discussion around this is suggested.

13. There are many different experiments squeezed into Fig 1. I would suggest to split it in more than one figure and include data currently shown in Supplementary.

Reviewer #2 (Remarks to the Author):

In this work, Ivanova and colleagues studied the mitotic DNA synthesis in budding yeast cells. The authors observed, in a substantial fraction of unperturbed yeast cells, an anaphase specific DNA synthesis for a subset of late-replicating regions, in particular those close to chromosome ends. They found that this ultra-late DNA synthesis might be related to the decrease in cyclin-Cdk activity during mitotic exit, and suggest that this anaphase DNA synthesis is an important mechanism to complete genome duplication before chromosome segregation.

GENERAL COMMENTS

The authors used both image analysis and genome-wide approach to study the mitotic DNA synthesis in budding yeast cells. They found that there is a large number of yeast cells (>40%) showing mitotic DNA synthesis under normal growth. They observed that this DNA synthesis is during anaphase but not during metaphase, which might be related to the difference level of cyclin-Cdk activity during these phases. This study provides an interesting observation on the process ensuring completion of genome duplication previous chromosome segregation in budding

yeast. However, some control experiments need to be added to draw a solid conclusion. Also, the relation between the current study with the previous reported works on MiDAS need to be discussed, and the detailed mechanisms need to be further investigated.

SPECIFIC COMMENTS:

Major comments:

- In figure 1a, the authors observed that G1 cells showed higher amount of EdU incorporation than metaphase cells. I wonder whether the different condensation levels of chromosome between G1 and metaphase cells might have some impact on the EdU incorporation/detection. Can the authors provide some control showing that this observation is not due to the problem of condensation? For example, the authors can label the newly replicated DNA by EdU within S phase then check the EdU incorporation signals within the cells at S, G2, metaphase, anaphase and next G1 phase. Without any technical bias, the authors should be able to observe similar signals within cells at different phases.

It will be better to compare directly the EdU incorporation level for the late mitotic cells (anaphase or telophase cells) and the metaphase cells as well.

- Previous studies have showed that exacerbated replication problems or spontaneous telomeric DNA damage responses in metaphase cells can lead to persistent DNA damage responses and DNA synthesis during mitosis. I wonder whether this can explain the association of the presence of RPA foci in anaphase cells and the delayed chromosome segregation observed in current study. The authors therefore need to investigate whether it presents telomere-specific DNA damage responses in their experimental systems, e.g. by measuring the colocalization of the gamma-H2AX signals with the telomere probes on metaphase chromosomes.

In addition, the authors should consider to use the EdU labelling together with telomere probes to check whether the anaphase DNA synthesis foci are located/enriched at sub-telomere regions.

- Intensive studies, from human and yeast, have revealed the roles and mechanisms of MiDAS and BIR on telomere maintenance (related with the ALT mechanism or not) and other late replicating regions such as CFSs, implicating these regions as DNA synthesis hot spots outside of S phase. The relation between the late mitotic DNA synthesis observed in current study with the mitotic DNA synthesis processes reported in previous studies need to be clearly discussed.

- To investigate the mechanism underlying the observed late mitotic DNA synthesis, it's important to check whether it is a conservative (BIR like) or a semiconservative (HR associated) DNA synthesis for the sub-telomere regions and the other late replicating regions, respectively. Do they belong to the same mechanism? Or different mechanisms are responsible for the DNA synthesis within different genomic contexts. How these mechanisms are associated with the Cdk inactivation during mitotic exit?

Also, for the results shown in Fig 3d-f, if separating the under-represented regions into two groups based on their distance to the telomeres, i.e. telomere-proximal and telomere-distal regions, and repeat the analyses, would similar results be observed for both groups?

- Based on the NGS data, over 40% of cells have lower metaphase copy number in the regions close to each telomere. This number seems very high. I wonder whether the authors can completely rule out the possibility that some technical bias might be able to be induced for the NGS of the sub-telomere regions at different stages of yeast mitotic cells. For example, the sucrose gradient centrifugation used to obtain the highest synchrony for DNA extraction might induce loss of sub-telomere regions due to DNA damage. To directly confirm the presence of DNA synthesis within these regions in late mitotic cells, the authors should also perform sequencing of the EdU/BrdU labelling DNA at the metaphase and anaphase/telophase cells. This kind of approach has been used in previous studies of MiDAS.

- The sub-telomere regions display higher mutations (including point mutations and Indel) than other genomic regions, either due to the DNA replication and/or bias gene conversion, which has been well documented in previous studies. Therefore, the results shown in Fig 4 a-c seems obvious, since the under-replicated regions enrich at sub-telomere regions. This part of study did not bring any new insights on the subject. In order to show that the mitosis DNA synthesis indeed contribute to the high mutation rate observed within the corresponding regions, mutation-accumulation experiment need to be performed at yeast strains with de-function of related genes involving in the process.

Minor comments:

- Page 2, Abstract: The term of "DNA replication" used in several places within the abstract for the mitotic DNA synthesis observed in this study. It will be better to replace them with "DNA synthesis" as explained in the Major Comments.

- Page 2, Abstract: "Replication during late mitosis correlates with elevated mutation rates, including copy number variation." to "DNA synthesis during late mitosis collocates with regions with elevated mutation rates, including copy number variation."

- Page 4, "we examined cells arrested in late anaphase with separated nuclei and high levels of mitotic cyclins by inactivation of the Mitotic Exit Network (MEN)". Ambiguous sentence. The cells arrested in late anaphase by inactivation of the MEM or the cells with high levels of mitotic cyclins by inactivation of the MEM?

- Page 4, "Moreover, time-lapse imaging of fluorescent loci in chromosome XII showed defects in the segregation of telomere-proximal regions in MEN mutant cells (Fig. S6)." What kinds of defects observed need to be described in more detail in the main text. In particular, as indicated in the legend of Fig. S6, "(B) The time of rDNA and subTel12R separation, relative to anaphase, is the same in WT and MEN mutants. (C) In MEN cells the subTel12R sister spots get closer to each other after their initial separation."

- Page 4, "timely bridge resolution after MEN reactivation required DNA polymerase delta, indicating that MEN-deficient bridges require DNA synthesis for their resolution (Fig. 2b)." In Fig 2b, the authors used *cdc15-as1 pol3-ts*. But as shown in Fig S4c, the impact of *pol3-ts* is much lower than that of *pol2-ts*. What will be the results for the *cdc15-as1 pol2-ts*?

- Page 6, "Consistent with this, over-expression of the mitotic cyclin CLB2 in S phase slowed completion of DNA replication following release from HU-induced cell cycle block..." then conclude that "Therefore, Cdk inactivation is required to complete replication of specific regions during mitotic exit, preventing the formation of stable chromatin bridges." Why is over-expression in S phase rather than in anaphase cells which is more directly demonstrate the result?

- Page 11, How many PCR cycles used during library preparation? Please indicate in the method.
- Page 12, "The percent of cells that have not yet replicated a given region of the genome (under-replication), is defined as $2 \times (1 - \text{CNR})$." Please provide more detail about the reason of such definition.
- Page 14, Fig 1 legend: "or arrested in metaphase at 25 °C and released at 37 °C to inactivate DNA replication." Add "of the indicated strains".
- Supplementary Figure 1: set the title in Bold.
- Supplementary Figure 2: please provide at least one image for each of the 6 categories in Fig S2c.
- Supplementary Figure 4: In Fig S4a, no difference was observed for the rad9 and rad9+HU. But in Fig S4b, the duration of chromatin bridges is shorter in rad9+HU than rad9 alone. Do you have any explanation for that?
- Supplementary Figure 8: in legend of Fig S8C indicated that "Control analysis for (B) using data from (13) with showing high copy-number around early-firing origins for sorted S-phase cells and non-sorted mid-log phase cells." But on the Fig S8C, it indicated "G1 arrested" and "M arrested", respectively. Is it a problem of legend?
- Supplementary Figure 8: in Fig S8B "mid-log phase" sample, lots of classes show a mean value <0. Do you have any explanation? Is it a problem of normalization?
- Supplementary Figure 9: Whether the length of under-presented regions for each chromosome end is correlated with length of the chromosome arm? Some chromosomes show almost no difference between the metaphase and telophase cells (such as chr II, VIII, IX, XI) but some others show very large difference (such as chr I, VI). Do you have any explanation about such variation?
- Supplementary Figure 11: in Fig S11B, 3rd row, for the Muller 2014 w303 50' dataset, a clear positive correlation was obtained in the figure on the left panel, but no correlation was obtained in the figure shown in the right panel. Is it normal?
- Supplementary Figure 13: only very slight difference was observed between the red and blue lines. Need to be clarified in the text.
- Supplementary Figure 14: please explain how the simulation was performed in the method section.

Reviewer #3 (Remarks to the Author):

Invanova, Mendoza et al. Budding Yeast Completes DNA Replication After Chromosome Segregation Begins

This is a (to me) mind-blowing report that asserts, with considerable and convincing evidence, that a substantial fraction of the budding yeast genome is under-replicated when cells enter metaphase, and only complete replication as cells complete mitosis. The evidence involves several techniques that corroborate their conclusions (including following live cells with fluorescent H2B

tagged cells for chromatin bridges; EdU pulse labeling of cells progressing through mitosis; RPA-GFP to mark ssDNA in different phases of the cell cycle; inhibition of CDK to allow completion of replication in metaphase-arrested cells; bioinformatics of deep sequencing to identify under-replicated regions in the genome, that tend to be telomeric). I find the entire story compelling.

Some parts of the narrative were a little difficult to follow, but not overly so.

Comments:

1. Vinton and Weinert 2017, provided some evidence that generally slowing the cell cycle stabilizes the genome. If the authors believe that story, and find it relevant, they might sight it. If not, I think its related but am unsure. Their supplemental figure 14 seemed related to the Vinton-Weinert report.

2. I was not clear on what the authors thought about the Rad9 controls. The authors might want to make explicit their point: the Rad9 checkpoint, then, gets activated but only if "S2" replication is blocked (by HU or DNAPol mutants)— though the Rad9 checkpoint is not activated by the slowed replication per se. Yet, the mid-anaphase delay that Bloom et al, ref 12, reported is somehow related to this S2 replication? If so, some of the S2 replication is triggering the Rad9 checkpoint, but???

3. I think it would be useful for the authors to state in their abstract how extensive this underreplication phenotype is; in 10 to 50% of the cells, to what fraction of the genome per cell cycle? Some sort of estimate would be useful for the reader.

4. These conclusions seem to run counter to the classical reciprocal shift experiments reported by Wood and Hartwell in 1982 (ref). The authors here should at least cite those observations, as readers deeply in the know of the Hartwell work might be confused. Those early studies reported apparently completed replication in nocodazole-arrested cells, as when nocodazole arreste cells (it was actually MBC drug) were shifted to HU, though only ~78% of the cells progressed to the next cell cycle. Those experiments were done in diploids, I believe, however, and these experiments in this manuscript are in haploids, perhaps accounting for the differences?

5. I found the tests of MEN controls and CDK completely convincing, as well as the bioinformatics of copy number and incomplete replication. Wow.
(huh...incomplete description...)

Data

6. Fig 1 A. What fraction of cells show EduG1+ phenotype? AU is arbitrary units..so % of phenotype not clear...

7. Nuclear division-- and bridge resolution and persistence of bridges are the same thing? Marked differently in fig 1C and D?

8. In Fib2B...is this bridge resolution before cytokinesis..so 60% of the cdc15 pol3 mutants did not resolve bridges before cytokinesis? And they do perform cytokinesis, but rather with unresolved bridges?

9. Fig 2C...I did not get this one....WT and cdc15 cells shown??? I only see mutant cells. It's a little tricky to look at: so fewer WT cells have RPA foci, viewed in cells as they go progress through anaphase, than cdc15 mutant cells (with inhibitor) as they progress though anaphase and arrest in telophase?

10. I thought that CDK levels were high in metaphase, and reduced to ~50% of that level in telophase? If so, that should be stated, and the model in Fig 4 adjusted to reflect that bump in

CDK levels.

11. Did the authors try the Dahmann-Diffley-Nasmyth experiment (Curr Biol. 1995 Nov 1;5(11):1257-69) experiment from 1995...nocodazole arrest, then induce Sic1 to lower CDK levels? That should allow EdU incorporation. At least cite this as a strategy that could be used: some in the know will think of that and wonder if the authors thought of it. I am not asking they do that too, just site it.

12. Fig 3B...what is the value on the Y axis for the region shown in green....maybe put the total kb of green above the green bit...gives the reader a feeling for how much of genome is underreplication. And this data is from arrest in metaphase, by what technique? Nocodazole?

13. Fig 3C...do any of those regions correspond to regions reported by Cha and Kleckner as breaking in mec1-4 mutants??

14, Fig 4A. genes that when deleted have high fitness are more often subject to underreplication...wow...and I am not sure exactly what I am to make of this correlation...that these genes are thus more likely under-expressed, so the cells are essentially hypomorphic for these genes??

We thank the reviewers for helping us improve the manuscript. Reviewer comments have been numbered and are in **bold black text**. Our replies are in blue.

Reviewer #1 (Remarks to the Author):

Ivanova and colleagues show that a small, but significant amount of DNA replication occurs out of S phase, in late stages of the budding yeast cell cycle. Authors show that cells in metaphase do not polymerize DNA, but DNA synthesis is resumed later in anaphase. This synthesis is necessary for proper nuclear division. High Cdk activity in metaphase is found to prevent completion of DNA replication while nuclear segregation requires DNA polymerases and MEN function. Deep sequencing analysis of G2/M cells identifies under-replicated regions in metaphase, which mainly comprise subtelomeric regions, transposable elements, fragile sites and G4-rich regions. Finally, an in silico analysis links late-replicating regions to high mutation rates, pointing to a possible evolutionary benefit for delaying replication until late mitosis.

The paper is convincing, some of the approaches are remarkable and data is technically sound. Overall, the manuscript supports the rather unexpected conclusion that Cdk activity in metaphase prevents completion of replication. This is an observation that breaks established models and should therefore merit its publication in Nature Communications. Although the manuscript does not have the mechanistic insight into how Cdk activity controls replication in mitosis, the observations are highly relevant and represent an important contribution to the fields of chromosome replication, segregation and cell cycle control.

My main criticism is that the manuscript is extremely concise. From my point of view, a longer version, including appropriate experiment descriptions and discussions, would make it more accessible.

We have lengthened the manuscript to make it more accessible. In particular, we have written a new introduction, expanded the results and discussion sections, and moved some supplementary figures to the main figure section.

There are some minor issues, mostly related to data shown in supplementary figures or not properly described in the main text:

1. It is not clear how authors differentiate EdU incorporation in DAPI-rich and DAPI-poor areas. Text is written as if DAPI-poor corresponds to the nucleolus. Have authors co-localized high EdU incorporation with nucleolar markers?

The reviewer is correct that DAPI-poor corresponds to the nucleolus. When budding yeast is treated with pheromone, the cell and nucleus are both polarized, with the spindle pole body, SPB (which is embedded in the nuclear envelope, NE) oriented towards the mating projection (shmoo), and the nucleolus located on the opposite side of the nucleus. The new **Figure 1c** in (former figure S1b) shows two examples of cells in which DAPI-poor regions are oriented away from the shmoo tip, surrounded by NE and co-localize with a nucleolar marker.

2. Also related to DAPI-poor areas, if the nucleolus is a major site for late mitotic replication, does polymerase inhibition in mitosis preferentially affect rDNA segregation?

To answer this question, we imaged nucleolar division in cells treated with HU during a metaphase block and release experiment (new **figure S5**, next page). We observed a statistically significant but very small delay in nucleolar segregation.

3. Fig 1a indicates that almost all cells increase EdU signal as they transit from metaphase to G1. Since not all chromosomes incorporate EdU, one would expect to localize EdU signal at specific sites (foci) rather than homogeneously dispersed throughout the nucleus/nucleolus.

We don't know why the EdU staining is homogenous, and agree that one possible reason for this is that EdU staining may reflect both DNA synthesis and increased nucleotide import at mitotic exit, as the reviewer suggests in the next point.

4. One final caveat about EdU incorporation is whether EdU is really polymerized into a new DNA strand or EdU uptake into the nucleus/nucleolus is simply upregulated as cells exit mitosis. Authors could address this by inactivating polymerases, as in Fig 1C.

As suggested by the reviewer, we inactivated DNA polymerase delta using an auxin-dependent degron (pol3-aid) in the same experimental setup used to test EdU incorporation. However, depletion of Pol3 in metaphase-arrested cells inhibits their progression into anaphase and therefore, lack of EdU incorporation may be due to inhibition of replication, inhibition of mitotic exit, or both (new supplementary figure 4, quantification [panel c] reproduced below).

This result is consistent with our time-lapse experiments, in which inhibition of DNA synthesis in mitosis delays or inhibits nuclear division (Figure 2). Therefore, although multiple lines of evidence indicate that a significant amount of DNA replication occurs in late mitosis, we cannot formally rule out the possibility that EdU uptake into the nucleus may be upregulated as cells exit mitosis. We have noted this in the results section:

“Nuclear EdU incorporation in G1 cells may reflect DNA synthesis in the nucleus and/or increased nuclear import of nucleotides. To test if EdU incorporation depends on DNA synthesis, we sought to inactivate DNA replication in metaphase arrested cells. Because the catalytic activity of DNA polymerase epsilon is not absolutely necessary for DNA replication (ref. 14,15) we chose to deplete the catalytic subunit of DNA polymerase delta using an auxin-inducible degron.

However, depletion of Pol3 in metaphase-arrested cells inhibited nuclear division (Fig. S4). This prevented us from formally demonstrating that nuclear EdU incorporation in G1 cells depends on DNA synthesis". (p. 5)

5. Authors should clarify at what stage in mitosis does DNA synthesis resume. Experiments shown in Fig S2 suggest it is not in anaphase (lowest level of EdU incorporation), while Fig 1 indicate it is not in metaphase. Then, it has to be after anaphase...? There is no example for "Stage 3" cells, also able to incorporate EdU. Are they in metaphase? Early anaphase?

We have added examples images of all the stages and expanded our explanation of this point (new figure S3). We interpret these results to mean EdU starts being incorporated in late anaphase or early G1:

"EdU incorporation, time-lapse imaging experiments and DNA copy number analysis all suggest that DNA synthesis stops or slows down during metaphase and resumes during late anaphase or early G1." (Discussion, p. 13)

6. Authors suggest the presence of a Rad9-dependent checkpoint, which delays nuclear division in response to replicative stress in M phase. This is nicely shown in Fig S4A, and described in just one sentence. However, there are some issues in the Fig S4B (duration of chromatin bridges) that require further clarifications. Specifically, authors should explain why bridges in rad9 cells (no HU) take longer to dissolve than in wt cells (Fig S4B). It is also quite puzzling that treatment with HU reduces duration of chromatin bridges in rad9 cells. In addition, the graph does not show big differences between chromatin bridge duration in wt and rad9 mutant cells when cells are treated with HU (Fig S4B). This does not fit with a model in which rad9 deletion advances nuclear division in HU (Fig S4A).

These are all good points. Because bridge lifetimes are relatively short (5-15 minutes), we repeated all *rad9* experiments (and relative controls) at 2-minute resolution and ensured that all cells were imaged under identical conditions on the same day. Specifically, we ran time-lapse experiments simultaneously, using multi-well microscopy chambers containing all strains to be compared. Two conclusions emerge from these new experiments, shown in the new figure 2a-c (graphs below). First, in untreated conditions (no HU), bridges take the same amount of time to dissolve between *rad9* and wt cells. Longer bridge lifetimes in *rad9* cells observed in our earlier experiments may have been due to batch effects. Second, and confirming our original results, bridges are long-lived in wt + HU but short-lived in *rad9* cells + HU, as seen below:

We agree that this does not fit a simple model in which *RAD9* deletion advances nuclear division in HU. If this were the case, one would expect that *rad9* cells should advance nuclear division in the presence of HU and display longer-lived bridges. We suggest an alternative interpretation: perhaps *RAD9* deletion leads to higher dNTP levels, which would facilitate S phase replication and reduce the defects associated with inhibition of M-phase replication. Accordingly, the Pasero lab demonstrated that CIN and checkpoint mutants have high dNTP levels that cause faster replication fork progression (although *rad9* was not investigated). We have included the following text in the results section:

“Interestingly, deletion of the *RAD9* checkpoint gene abolished nuclear division delays and chromatin bridge formation in response to challenges in DNA synthesis during mitosis (**Fig. 2b-c**). This may indicate that Rad9 responds to DNA damage during early anaphase, as suggested by a previous study¹⁷. Alternatively, *rad9Δ* strains may be less sensitive to perturbation of mitotic DNA synthesis, for example due to higher dNTP levels that accelerate fork progression during the preceding S-phase¹⁸. Consistent with this possibility, chromosome instability mutants and checkpoint-defective *rad53* cells have elevated dGTP levels^{18,19}, although to our knowledge, whether dNTP levels are higher in *rad9* mutants than in wild-type cells is not known.” (p. 6)

7. Related to point 5 and 6, it is also unclear at what stage Rad9 arrests cells. Is it before anaphase entry? This is relevant, as a checkpoint responding to defects in DNA polymerization should become active after DNA polymerization has been attempted.

See our answer to point 6.

8. Fig S4C and D. It seems that polepsilon and GINS promote chromatin bridge resolution and nuclear division in mitosis, while polalpha and poldelta do not. Although not all ts alleles used may affect DNA replication to the same extent, the (possible) differential contribution of DNA polymerases should be discussed in the text.

The main conclusion we draw from these experiments is that challenging DNA synthesis during mitosis by multiple mechanisms (HU addition or *ts* mutations) cause similar phenotypes (delays in nuclear division). We prefer not to over-interpret differences between conditions based on the available data, given the many possible sources of variation: not only penetrance of specific *ts* alleles, but also variation in the time necessary to inactivate different *ts* proteins. For instance, *pol2* (but not *pol3*) cells had severe defects in nuclear division after release from metaphase (**Fig. 2d-e**), whereas both mutants showed severe defects in chromosome bridge resolution in late anaphase (**Fig. 5c**). See also our reply to point 12 of reviewer 2. With this in mind, we have added the following text to the results:

“The severity of nuclear and cell division delays varied among *ts* mutants, with *pol2* mutant cells displaying the strongest defects, *psf2* and *pol3* displayed relatively mild delays and *pol1* showing no effect (**Fig. 2c-e**). These differences might be due to polymerase-specific functions or to variations in the penetrance and inactivation kinetics of different *ts* alleles”. (p. 6)

9. Figure S4 legend is incomplete and difficult to follow. Also, there are no references in the main text to some parts in Fig S4.

The legend has been rewritten.

10. Figure 2B. There is no control for no *cdc15-as* reactivation. Cells are monitored for extended periods. Authors should discard that changes observed during 1NA-PP1 washout are a matter of time, not dependent on Cdc15 reactivation.

To verify that Cdc15-*as1* does not reactivate after a prolonged time in the presence of the 1NA-PP1, we quantified bridge resolution in *cdc15-as1* cells treated with the inhibitor for 3 hours, and then imaged for another 3 hours without inhibitor washout. Under these conditions, 87% of cells remain arrested in telophase with intact chromatin bridges. These results were included in supplementary figure 7. We have moved them to the main manuscript (see Figure 5a, see below).

11. Fig S13. Over-expression of CLB2. There are not big differences in replication timing between cultures expressing or not CLB2. An effect of sustained high mitotic Cdk activity would be better addressed by deep sequencing or accumulation of anaphase bridges, probably more sensitive than FACS.

We agree, and have removed this figure.

12. The observation that Cdk activity prevents completion of DNA replication in mitosis is unexpected, given its role in promoting DNA synthesis in S phase. The real question here is how Cdk activity blocks completion of replication. Although authors speculate about direct Cdk inhibition of DNA synthesis (fork pausing during metaphase to prevent chromosome pulverization), an indirect effect is also possible. For example, there are reports linking Cdc14-dependent Cdk downregulation to RNA polymerase transcription and chromosome segregation. As transcription can block replication fork progression, a modification of the discussion around this is suggested.

A mechanism for how M-phase Cdk blocks completion of replication has been recently proposed: when mitotic CDK is used to drive interphase *Xenopus* egg extracts into a mitotic state, the replicative CMG helicase undergoes ubiquitylation and is removed from chromatin by the Cdc48 ATPase (Deng et al., Mol Cell 2019). A similar mechanism may operate in budding yeast. However, we agree with the reviewer that indirect effects such as transcription / replication conflicts may also play a role. We now include these points in the discussion, as follows:

“Although high Cdk levels are compatible with bulk DNA replication in fission yeast², chromatin replication is inhibited in *Xenopus* mitotic extracts owing to Cdk-dependent replisome disassembly^{30,31}. Perhaps high CDK levels block replication during budding yeast mitosis through a similar mechanism, pausing replication during metaphase to prevent damage and chromosome pulverisation^{32,33}. The anaphase state may also promote completion of replication through alternative mechanisms. For instance, the Cdc14 phosphatase inhibits transcription during yeast anaphase, and this has been proposed to facilitate

chromosome condensation³⁴. As transcription can block replication fork progression, inhibition of transcription may not only promote chromosome condensation but also facilitate completion of replication during anaphase". (p. 13)

13. There are many different experiments squeezed into Fig 1. I would suggest to split it in more than one figure and include data currently shown in Supplementary.

We agree with this. The new Figure 1 now contains all EdU related data from the previous version (old Fig. 1a and S1). The new Figure 2 contains fluorescence time-lapse microscopy data (old Figure 1b-d and S4).

Reviewer #2 (Remarks to the Author):

In this work, Ivanova and colleagues studied the mitotic DNA synthesis in budding yeast cells. The authors observed, in a substantial fraction of unperturbed yeast cells, an anaphase specific DNA synthesis for a subset of late-replicating regions, in particular those close to chromosome ends. They found that this ultra-late DNA synthesis might be related to the decrease in cyclin-Cdk activity during mitotic exit, and suggest that this anaphase DNA synthesis is an important mechanism to complete genome duplication before chromosome segregation.

GENERAL COMMENTS

The authors used both image analysis and genome-wide approach to study the mitotic DNA synthesis in budding yeast cells. They found that there is a large number of yeast cells (>40%) showing mitotic DNA synthesis under normal growth. They observed that this DNA synthesis is during anaphase but not during metaphase, which might be related to the difference level of cyclin-Cdk activity during these phases. This study provides an interesting observation on the process ensuring completion of genome duplication previous chromosome segregation in budding yeast. However, some control experiments need to be added to draw a solid conclusion. Also, the relation between the current study with the previous reported works on MiDAS need to be discussed, and the detailed mechanisms need to be further investigated.

We have added controls, mentioned potential caveats in the interpretation of some results, and discussed the relationship with MiDAS, as detailed below.

SPECIFIC COMMENTS:

Major comments:

1 - In figure 1a, the authors observed that G1 cells showed higher amount of EdU incorporation than metaphase cells. I wonder whether the different condensation levels of chromosome between G1 and metaphase cells might have some impact on the EdU incorporation/detection. Can the authors provide some control showing that this observation is not due to the problem of condensation? For example, the authors can label the newly replicated DNA by EdU within S phase then check the EdU incorporation signals within the cells at S, G2, metaphase, anaphase and next G1 phase. Without any technical bias, the authors should be able to observe similar signals within cells at

different phases. It will be better to compare directly the EdU incorporation level for the late mitotic cells (anaphase or telophase cells) and the metaphase cells as well.

We thank the reviewer for the suggestion. However, we think the proposed control is not necessary because there is no correlation between EdU levels and chromosome condensation. In budding yeast, chromosome condensation is low in G1, intermediate in metaphase, and reaches its maximum in anaphase (see for example Neurohr et al., *Science* 2011). In contrast, nuclear EdU incorporation is highest in G1 and late anaphase (when condensation is at its lowest and highest, respectively), and lowest in metaphase and early anaphase (medium and high condensation). Thus, it is unlikely that condensation levels have an impact in EdU incorporation or detection.

2- Previous studies have showed that exacerbated replication problems or spontaneous telomeric DNA damage responses in metaphase cells can lead to persistent DNA damage responses and DNA synthesis during mitosis. I wonder whether this can explain the association of the presence of RPA foci in anaphase cells and the delayed chromosome segregation observed in current study. The authors therefore need to investigate whether it presents telomere-specific DNA damage responses in their experimental systems, e.g. by measuring the colocalization of the gamma-H2AX signals with the telomere probes on metaphase chromosomes.

In addition, the authors should consider to use the EdU labelling together with telomere probes to check whether the anaphase DNA synthesis foci are located/enriched at sub-telomere regions.

To test for DNA damage in metaphase-arrested cells, we measured Rad53 phosphorylation after prolonged depletion of Cdc20 and found none (see below, Fig. S2).

We did not attempt EdU labeling together with telomere probes. As noted by rev. 1, the EdU staining is diffuse throughout the nucleus and enriched in the nucleolus; co-localization with telomere foci would not be conclusive.

3- Intensive studies, from human and yeast, have revealed the roles and mechanisms of MiDAS and BIR on telomere maintenance (related with the ALT mechanism or not) and other late replicating regions such as CFSS, implicating these regions as DNA synthesis hot spots outside of S phase. The relation between the late mitotic DNA synthesis observed in current study with the mitotic DNA synthesis processes reported in previous studies need to be clearly discussed.

We agree, and have included a discussion on the role of MiDAS and BIR (Discussion, last paragraph). We have also included new experiments suggesting that Pol32 (which is required for BIR) promotes resolution of chromatin bridges in anaphase (see next point).

4- To investigate the mechanism underlying the observed late mitotic DNA synthesis, it's important to check whether it is a conservative (BIR like) or a semiconservative (HR associated) DNA synthesis for the sub-telomere regions and the other late replicating regions, respectively. Do they belong to the same mechanism? Or different mechanisms are responsible for the DNA synthesis within different genomic contexts. How these mechanisms are associated with the Cdk inactivation during mitotic exit?

Although a detailed mechanism of DNA synthesis during mitosis is outside of the scope of our study, we addressed the possible contribution of BIR to the resolution of chromatin bridges in late mitosis. These experiments suggest that Pol32, which is required for BIR, plays a role in the resolution of chromatin bridges after MEN reactivation. In contrast, Rad51 is not required for bridge resolution. Note however that Pol32-independent bridge resolution mechanisms must exist, since a relatively higher fraction of *pol32Δ* cells was competent in bridge resolution compared to much more severe defects after inactivation of Pol3 or Pol2.

These results are shown in new figure 5c below:

5- Also, for the results shown in Fig 3d-f, if separating the under-represented regions into two groups based on their distance to the telomeres, i.e. telomere-proximal and telomere-distal regions, and repeat the analyses, would similar results be observed for both groups?

The extent of under-replication is higher in late replicating regions across whole genome, regardless of their proximity to the telomeres (Figure S12a, see right). This strengthens our confidence in the result; it would be surprising if non-subtelomeric under-replicated regions were not late-replicating.

Additionally, we observe that transposable elements and fragile sites are under-replicated specifically in telomere-distant regions, suggesting that there is a certain intrinsic property of these elements that causes them to be under-replicated, independent of chromosomal location (Figure S12b, right). On the contrary, G4-rich regions are under-replicated only in subtelomeric regions, suggesting that G4-rich regions are under-replicated mainly due to their positioning along chromosomes.

6- Based on the NGS data, over 40% of cells have lower metaphase copy number in the regions close to each telomere. This number seems very high. I wonder whether the authors can completely rule out the possibility that some technical bias might be able to be induced for the NGS of the sub-telomere regions at different stages of yeast mitotic cells. For example, the sucrose gradient centrifugation used to obtain the highest synchrony for DNA extraction might induce loss of sub-telomere regions due to DNA damage. To direct confirming the presence of DNA synthesis within these regions in late mitotic cells, the authors should also perform sequencing of the EdU/BrdU labelling DNA at the metaphase and anaphase/telophase cells. This kind of approach has been used in previous studies of MiDAS.

The sucrose gradient centrifugation was used in NGS experiments in figure 6a-e, but not on those in figure 6f that gave essentially identical results. However, we take the reviewers' point that validation of NGS results with orthogonal data would be useful. We tried BrdU-seq as suggested, although we anticipated technical difficulties with this approach, since our previous results predicted that only low levels of DNA synthesis occur during late mitosis. Indeed, although we did identify an enrichment of subtelomeric and rDNA sequences in some BrdU-seq samples (consistent with our NGS data) we failed to obtain robust results in all replicates, perhaps due to the extremely low amounts of DNA recovered in IPs from G1-arrested cells. We have therefore decided not to include these results, and cannot rule out that technical bias may be influencing the DNA sequencing data. On the other hand, since both EdU incorporation and time-lapse microscopy data support our model, and since the numbers from HU and pol2 inhibition (Fig 2) roughly match 40% from microscopy, we think that the interpretation that DNA synthesis during late mitosis is important for chromosome segregation is probably correct.

7- The sub-telomere regions display higher mutations (including point mutations and Indel) than other genomic regions, either due to the DNA replication and/or bias gene conversion, which has been well documented in previous studies. Therefore, the results shown in Fig 4 a-c seems obvious, since the under-replicated regions enrich at sub-

telomere regions. This part of study did not bring any new insights on the subject. In order to show that the mitosis DNA synthesis indeed contribute to the high mutation rate observed within the corresponding regions, mutation-accumulation experiment need to be performed at yeast strains with de-function of related genes involving in the process. We agree that increased mutation rates in sub-telomeric regions was previously known. However, the mechanism behind higher mutation rates in these regions remained unclear. Our results suggested that late replication and/or segregation could be causing higher mutations in these regions. We envisage two non-mutually exclusive mechanisms: (1) replication pathways used in late mitosis could be intrinsically more error-prone than in S-phase, and/or (2) chromatin bridges could be exposed to cytokinesis-dependent damage. The first possibility is supported by our finding that Pol32 is involved in anaphase bridge resolution (new figure 5c). Pol32 is involved in error-prone BIR.

To explore the second possibility, we inserted a *URA3* marker at various distances from a telomere, and determined how often expression of this marker was lost in wild type and in cells delayed in cytokinesis (new Figure 7d on the right). We find that as expected, (1) marker loss frequency increases with proximity to the telomere and (2) further increases when DNA replication origin firing is challenged in *sic1Δ* strains. Interestingly, delaying cytokinesis (in a *cyk3Δ* mutant) reduced *URA3* mutation specifically in telomere-proximal loci, but not in telomere-distal positions. These results support the hypothesis that the increased mutation rate of late-replicating, late-segregating chromosome regions is due, at least in part, to cytokinesis-dependent damage.

Minor comments:

8- Page 2, Abstract: The term of “DNA replication” used in several places within the abstract for the mitotic DNA synthesis observed in this study. It will be better to replace them with “DNA synthesis” as explained in the Major Comments.

We have replaced “DNA replication” with “DNA synthesis” throughout the manuscript, except in certain contexts where we used “replication” in the context of our interpretation of the data.

9- Page2, Abstract: “Replication during late mitosis correlates with elevated mutation rates, including copy number variation.” to “DNA synthesis during late mitosis collocates with regions with elevated mutation rates, including copy number variation.”

We have changed the text as suggested.

10- Page 4, “we examined cells arrested in late anaphase with separated nuclei and high levels of mitotic cyclins by inactivation of the Mitotic Exit Network (MEN)”. Ambiguous sentence. The cells arrested in late anaphase by inactivation of the MEM or the cells with high levels of mitotic cyclins by inactivation of the MEM?

We have replaced the text as follows: “we examined cells defective in the Mitotic Exit Network (MEN), a kinase cascade pathway required to inactivate Cdk at the end of mitosis. Temperature-sensitive MEN mutants arrest in late anaphase at the restrictive temperature, with separated nuclei and high levels of mitotic cyclins” (p. 7).

11- Page 4, “Moreover, time-lapse imaging of fluorescent loci in chromosome XII showed defects in the segregation of telomere-proximal regions in MEN mutant cells (Fig. S6).” What kinds of defects observed need to be described in more detail in the main text. In particular, as indicated in the legend of Fig. S6, “(B) The time of rDNA and subTel12R separation, relative to anaphase, is the same in WT and MEN mutants. (C) In MEN cells the subTel12R sister spots get closer to each other after their initial separation.”.

We have moved this figure to the main manuscript (Fig 4b-d) and described in more detail in the main text.

12- Page 4, “timely bridge resolution after MEN reactivation required DNA polymerase delta, indicating that MEN-deficient bridges require DNA synthesis for their resolution (Fig. 2b).” In Fig 2b, the authors used *cdc15-as1 pol3-ts*. But as shown in Fig S4c, the impact of *pol3-ts* is much lower than that of *pol2-ts*. What will be the results for the *cdc15-as1 pol2-ts*?

We performed the experiment suggested by the reviewer. As shown in the new **Figure 5c** (see point 4), *cdc15-as1 pol3-ts* double mutants have a strong defect in bridge resolution, comparable to that of *cdc15-as1 pol2-ts* strains. Thus, Pol2 and Pol3 are required for chromosome segregation after release from a Cdc15 block.

Interestingly, and as noted by the reviewer, our data suggested only a minor role for Pol3 in chromosome segregation after release from a metaphase block. What could explain the different requirement for Pol3 in the two experiments? One possibility is slow inactivation of the Pol3-ts protein after shift to the restrictive temperature. Two observations support this interpretation. First, acute depletion of Pol3 with an auxin-dependent degron (*pol3-aid*) impairs nuclear division more severely than temperature shift of *pol3-ts* (new **figure S4**). This opens the possibility that the Pol3-ts protein is partially active in the first anaphase following release from a metaphase block. Second, cells re-enter the cycle faster when released from a metaphase block than they do from a Cdc15 block. Indeed, in our hands *MET3pr-CDC20* cells enter anaphase approximately 30 minutes after methionine washout, whereas *cdc15-as1* cells exit mitosis 90 minutes after NAPP1 washout. Because in both experiments, we shifted cells to the restrictive temperature at the same time relative to washout of the inhibitors, Pol3 activity may be higher during the first anaphase in cells released from a metaphase block than in cells released from a Cdc15 block. In any case, based on our results we conclude that both Pol2 and Pol3 play important roles in chromosome segregation during late mitosis.

13- Page 6, “Consistent with this, over-expression of the mitotic cyclin CLB2 in S phase slowed completion of DNA replication following release from HU-induced cell cycle block...” then conclude that “Therefore, Cdk inactivation is required to complete replication of specific regions during mitotic exit, preventing the formation of stable chromatin bridges.” Why is over-expression in S phase rather than in anaphase cells which is more directly demonstrate the result?

We agree with comments from reviewer 1 concerning this experiment, and have decided to remove this figure.

14- Page 11, How many PCR cycles used during library preparation? Please indicate in the method.

The library prep was PCR-free. This information has been added to the methods.

15- Page 12, “The percent of cells that have not yet replicated a given region of the genome (under-replication), is defined as $2*(1-CNR)$.” Please provide more detail about the reason of such definition.

We have added a detailed explanation in a greatly expanded "Quantification of under-replication" section in the methods.

16- Page 14, Fig 1 legend: “or arrested in metaphase at 25 °C and released at 37 °C to inactivate DNA replication.” Add “of the indicated strains”.

This has been corrected.

17- Supplementary Figure 1: set the title in Bold.

This has been corrected.

18- Supplementary Figure 2: please provide at least one image for each of the 6 categories in Fig S2c.

This has been added (now Figure S3).

19- Supplementary Figure 4: In Fig S4a, no difference was observed for the rad9 and rad9+HU. But in Fig S4b, the duration of chromatin bridges is shorter in rad9+HU than rad9 alone. Do you have any explanation for that?

This has been addressed. See our answer to rev 1, point 6.

20- Supplementary Figure 8: in legend of Fig S8C indicated that “Control analysis for (B) using data from (13) with showing high copy-number around early-firing origins for sorted S-phase cells and non-sorted mid-log phase cells.” But on the Fig S8C, it indicated “G1 arrested” and “M arrested”, respectively. Is it a problem of legend?

It was a mistake in the legend. We have fixed it.

21- Supplementary Figure 8: in Fig S8B “mid-log phase” sample, lots of classes show a mean value <0. Do you have any explanation? Is it a problem of normalization?

We believe this is a typo, and the reviewer means < 1 . All values in this panel are positive. This behaviour (median (red line) > 1 for early and < 1 for late) in mid-log phase cells is expected. In a mid-log phase population, some cells will be in G1, some cells will have replicated only early origins, and some cells will have replicated early and mid, and some cells will have completed replication. Therefore, the expected copy-number of the 1-kb regions around each ARS is early-firing $>$ mid-firing $>$ late-firing. This is observed behaviour in mid-log phase cells, and, more strongly, in S-phase cells. All data in B is from Alvino. et al. Mol. Cell. Biol. 2007.

22- Supplementary Figure 9: Whether the length of under-presented regions for each chromosome end is correlated with length of the chromosome arm? Some chromosomes show almost no difference between the metaphase and telophase cells (such as chr II, VIII, IX, XI) but some others show very large difference (such as chr I, VI). Do you have any explanation about such variation?

We address these issues with a new figure (see next page) and a new Discussion section:

“Why the level of mitotic under-replication varies between chromosomes, and

between metaphase and anaphase, remain open questions. Intriguingly, the length of under-replication and chromosome arm length are negatively correlated ($r=-0.65$) (**Figure S14a**). One possible explanation for this correlation is that longer chromosome arms have more potential origins of replication, thus decreasing their probability of under-replication. Indeed, arms with longer under-replicated regions have fewer origins (**Figure S14b**). However, within subtelomeric regions (75 kb) both the absolute number and the density of origins (number of origins divided by the length of the chromosome arm) are positively correlated with length of under-replication; i.e. arms with more under-replicated DNA have more origins in the under-replicated region (**Figure S14 c,d**). Moreover, arms with the highest under-replication in metaphase relative to telophase tend to be shorter and have a higher ARS density (**Figure S14e-h**). That being said, firing of subtelomeric origins is highly context-dependent and variable between cells, and it is likely that not all origins fire in all cells. Therefore, the precise causal relation between under-replication, arm length, and origin density, if any, remains to be discovered” (p.14)

23- Supplementary Figure 11: in Fig S11B, 3rd row, for the Muller 2014 w303 50' dataset, a clear positive correlation was obtained in the figure on the left panel, but no correlation was obtained in the figure shown in the right panel. Is it normal?

We believe that this is because there were no replicates in that experiment. As opposed to the other experiments (see titles of each subplot). We analyzed all of the publicly available data, and we therefore show all the results, but we wouldn't put too much faith in, nor spend too much time thinking about, an experiment that hasn't been repeated.

24- Supplementary Figure 13: only very slight difference was observed between the red and blue lines. Need to be clarified in the text.

We agree with comments from reviewer 1 concerning this experiment, and have decided to remove this figure.

25- Supplementary Figure 14: please explain how the simulation was performed in the method section.

A detailed description was added in the methods.

Reviewer #3 (Remarks to the Author):

Invanova, Mendoza et al. Budding Yeast Completes DNA Replication After Chromosome Segregation Begins

This is a (to me) mind-blowing report that asserts, with considerable and convincing evidence, that a substantial fraction of the budding yeast genome is under-replicated when cells enter metaphase, and only complete replication as cells complete mitosis. The evidence involves several techniques that corroborate their conclusions (including following live cells with fluorescent H2B tagged cells for chromatin bridges; EdU pulse labeling of cells progressing through mitosis; RPA-GFP to mark ssDNA in different phases of the cell cycle; inhibition of CDK to allow completion of replication in metaphase-arrested cells; bioinformatics of deep sequencing to identify under-replicated regions in the genome, that tend to be telomeric). I find the entire story compelling.

Some parts of the narrative were a little difficult to follow, but not overly so.

We have lengthened the manuscript to make it more accessible. In particular, we have written a new introduction, expanded the results and discussion sections, and moved some supplementary figures to the main figure section.

Comments:

1. Vinton and Weinert 2017, provided some evidence that generally slowing the cell cycle stabilizes the genome. If the authors believe that story, and find it relevant, they might sight it. If not, I think its related but am unsure. Their supplemental figure 14 seemed related to the Vinton-Weinert report.

Thank you for reminding us about this paper. We have added this to the discussion.

2. I was not clear on what the authors thought about the Rad9 controls. The authors might want to make explicit their point: the Rad9 checkpoint, then, gets activated but only if “S2” replication is blocked (by HU or DNAPol mutants)— though the Rad9 checkpoint is not activated by the slowed replication per se. Yet, the mid-anaphase delay that Bloom et al, ref 12, reported is somehow related to this S2 replication? If so, some of the S2 replication is triggering the Rad9 checkpoint, but???

For a detailed discussion of Rad9-related issues, see our reply to rev. 1 point 6. In summary, a Rad9 checkpoint that detects attempted DNA replication in anaphase may exist, but an alternative explanation for our results is that the *RAD9* deletion leads to higher dNTP levels, which would facilitate S phase replication and reduce the defects associated with inhibition of M-phase replication. We have included the following text in the results section:

“Interestingly, deletion of the *RAD9* checkpoint gene abolished nuclear division delays and chromatin bridge formation in response to challenges in DNA synthesis during mitosis (**Fig. 2b-c**). This may indicate that Rad9 responds to DNA damage during early anaphase, as suggested by a previous study¹⁷. Alternatively, *rad9Δ* strains may be less sensitive to perturbation of mitotic DNA synthesis, for example due to higher dNTP levels that accelerate fork progression during the preceding S-phase¹⁸. Consistent with this possibility, chromosome instability mutants and checkpoint-defective *rad53* cells have elevated dGTP levels^{18,19}, although to our

knowledge, whether dNTP levels are higher in *rad9* mutants than in wild-type cells is not known.” (p. 6)

3. I think it would be useful for the authors to state in their abstract how extensive this underreplication phenotype is; in 10 to 50% of the cells, to what fraction of the genome per cell cycle? Some sort of estimate would be useful for the reader.

We agree, and have included this information in the abstract and introduction.

4. These conclusions seem to run counter to the classical reciprocal shift experiments reported by Wood and Hartwell in 1982 (ref). The authors here should at least cite those observations, as readers deeply in the know of the Hartwell work might be confused. Those early studies reported apparently completed replication in nocodazole-arrested cells, as when nocodazole arrested cells (it was actually MBC drug) were shifted to HU, though only ~78% of the cells progressed to the next cell cycle. Those experiments were done in diploids, I believe, however, and these experiments in this manuscript are in haploids, perhaps accounting for the differences?

We thank the reviewer for pointing out the Wood and Hartwell paper (PMID: 6752153), which we now cite. Our findings actually agree with the reciprocal shift experiments in that classic study. Specifically, both our results and Wood/Hartwell’s find that approximately 80% of cells progress to the next cycle when released from a metaphase arrest into HU-containing medium. Likewise, *pol1-ts* and *pol3-ts* mutants (termed *cdc2-1* and *cdc17-1* in the Wood and Hartwell paper) have only a mild division defect when shifted to the restrictive temperature after metaphase arrest. As mentioned in point 8 of reviewer 1, this may reflect differential contribution of DNA polymerases to anaphase replication, or differences in penetrance of the alleles examined. We did not examine the other DNA replication mutants tested by Wood and Hartwell (*cdc6*, *cdc7* and *cdc9*) but we note that *CDC6* and *CDC7* are required for initiation of DNA synthesis, consistent with the idea that replication elongation, but not initiation, promotes completion of replication during late mitosis.

5. I found the tests of MEN controls and CDK completely convincing, as well as the bioinformatics of copy number and incomplete replication. Wow. (huh...incomplete description...)

We are pleased that the reviewer finds these experiments convincing.

Data

6. Fig 1 A. What fraction of cells show EduG1+ phenotype? AU is arbitrary units..so % of phenotype not clear...

Units correspond to mean nuclear fluorescence intensity in single cells, after subtraction of cytoplasmic background. All cells with EdU incorporation in mitochondria were quantified, to control for cellular uptake and labeling of EdU. We have repeated this experiment in a manner that allows us to quantify all cells, by adding ethidium bromide shortly before EdU, thus reducing the mitochondrial DNA signal. This is now shown in figure S1, see next page.

7. Nuclear division-- and bridge resolution and persistence of bridges are the same thing? Marked differently in fig 1C and D?

Nuclear division is defined as the time between release from metaphase and resolution of chromatin bridges (final DNA segregation). Bridge duration is defined as the time between anaphase onset (nuclear elongation) and bridge resolution. This has been added to the figure 2 legend.

8. In Fib2B...is this bridge resolution before cytokinesis..so 60% of the *cdc15 pol3* mutants did not resolve bridges before cytokinesis? And they do perform cytokinesis, but rather with unresolved bridges?

Yes. A new explanation has been added to the results section:

“ (...) Inactivation of DNA polymerase delta or epsilon prevented bridge resolution during anaphase. Indeed, MEN reactivation in *pol3-ts* and *pol2-ts* cells caused bridge disappearance only during or after actomyosin ring contraction. This suggests that in the absence of DNA polymerase function, bridges are damaged by cytokinesis (“cut phenotype”), and that MEN-deficient bridges require DNA synthesis during anaphase for their timely resolution” (p. 9).

9. Fig 2C...I did not get this one....WT and *cdc15* cells shown??? I only see mutant cells. It's a little tricky to look at: so fewer WT cells have RPA foci, viewed in cells as they progress through anaphase, than *cdc15* mutant cells (with inhibitor) as they progress through anaphase and arrest in telophase?

We only show a mutant cell in the panel. The graph quantifies the % of cells that start anaphase with RPA foci for wild type and *cdc15* mutants. As noted in the legend, only the first 20 minutes of anaphase are considered for this analysis. We have rewritten this section to improve clarity.

10. I thought that CDK levels were high in metaphase, and reduced to ~50% of that level in telophase? If so, that should be stated, and the model in Fig 4 adjusted to reflect that bump in CDK levels.

Cells defective in *Cdc15* arrest in telophase with about 50% of Cyclin B levels relative to metaphase, but this is not necessarily the level of CycB present during a normal telophase (or the level of CDK activity, which also depends on other factors such as Sic1). We depict a

gradual decrease in CDK activity as cells exit mitosis in Fig 4 (now Fig. 8) only for illustrative purposes, and this figure is not intended to accurately depict CDK kinetics during mitotic exit.

11. Did the authors try the Dahmann-Diffley-Nasmyth experiment (Curr Biol. 1995 Nov 1;5(11):1257-69) experiment from 1995...nocodazole arrest, then induce Sic1 to lower CDK levels? That should allow EdU incorporation. At least cite this as a strategy that could be used: some in the know will think of that and wonder if the authors thought of it. I am not asking they do that too, just site it.

If we understand correctly, the proposed experiment would test the hypothesis that reducing CDK activity in metaphase inhibits DNA synthesis. Although we did not do the experiment as proposed, we performed two other experiments that address the same question.

- First, we arrested cells in metaphase (by Cdc20 depletion), inactivated CDK (with a *cdc28-as1* mutant), and tested if DNA synthesis was completed in subtelomeric regions by DNA sequencing (figure 6f, formerly 3f).
- - Second, we arrested cells in late mitosis with high CDK levels (by Cdc15 inactivation), inactivated CDK (with a *cdc28-ts* mutant) and used time-lapse microscopy to test if under-replicated DNA bridges are resolved (figure 6g, formerly 3g).

Since these two experiments indicate that mitotic DNA synthesis resumes when CDK activity is reduced, and given the caveats associated with EdU incorporation, we have decided not to perform the suggested experiment.

12. Fig 3B...what is the value on the Y axis for the region shown in green....maybe put the total kb of green above the green bit...gives the reader a feeling for how much of genome is underreplication.

Yes, that is correct. We have made this clearer in the figure legend.

And this data is from arrest in metaphase, by what technique? Nocodazole?

MET3pr-CDC20. We have made this clearer in the figure legend.

13. Fig 3C...do any of those regions correspond to regions reported by Cha and Kleckner as breaking in *mec1-4* mutants??

There does appear to be more overlap than one would expect by chance. That said, Cha and Kleckner did not make their raw data available, so we cannot say for sure. Looking at Figure S2 in their paper, Breakpoints I, V, VI are subtelomeric and show under-replication. Breakpoint II is not subtelomeric and also shows under-replication. Breakpoint III shows lower under-replication than you expect, but this might be also a sign of fragile site where DNA could break easily. Breakpoint IV is not under-replicated.

14, Fig 4A. genes that when deleted have high fitness are more often subject to underreplication...wow...and I am not sure exactly what I am to make of this correlation...that these genes are thus more likely under-expressed, so the cells are essentially hypomorphic for these genes??

This panel shows that the deletion of genes that are replicated late in mitosis does not affect fitness; these genes are dispensable for growth in rich media. It is true that these genes also tend to have lower expression, but replication timing does not seem to have large effects on expression in wild-type yeast [Voichek Bar-Ziv Barkai, Science 2016].

Reviewers' comments:

Reviewer #1 (Remarks to the Author):

The revised manuscript by Ivanova and colleagues provides additional experimental data and explanations for all the issues I raised in the previous version. I particularly like the the alternative hypothesis (elevated dNTP levels during the previous S phase) proposed to explain the apparent paradox of the rad9- mutant.

I am very happy to endorse publication of this excellent manuscript. It is a fascinating analysis about how budding yeast cells complete DNA replication at an unexpected time in the cell cycle.

One minor comment in Figure S3 Legend: 'Cells were growing in mid-log phase were treated...'

Reviewer #2 (Remarks to the Author):

I'm glad that, in this revised version, the authors addressed most of the comments raised by the reviewers and improved significantly their manuscript by including additional results and figures, and the authors also made more attention on potential caveats in the interpretation of the results. I believe that this study will provide interesting results for the community, while there are still several important issues need to be addressed. These points are outlined below:

1. In the response to the author:

"In figure 1a, the authors observed that G1 cells showed higher amount of EdU incorporation than metaphase cells. I wonder whether the different condensation levels of chromosome between G1 and metaphase cells might have some impact on the EdU incorporation/detection. Can the authors provide some control showing that this observation is not due to the problem of condensation? For example, the authors can label the newly replicated DNA by EdU within S phase then check the EdU incorporation signals within the cells at S, G2, metaphase, anaphase and next G1 phase. Without any technical bias, the authors should be able to observe similar signals within cells at different phases. It will be better to compare directly the EdU incorporation level for the late mitotic cells (anaphase or telophase cells) and the metaphase cells as well.

We thank the reviewer for the suggestion. However, we think the proposed control is not necessary because there is no correlation between EdU levels and chromosome condensation. In budding yeast, chromosome condensation is low in G1, intermediate in metaphase, and reaches its maximum in anaphase (see for example Neurohr et al., Science 2011). In contrast, nuclear EdU incorporation is highest in G1 and late anaphase (when condensation is at its lowest and highest, respectively), and lowest in metaphase and early anaphase (medium and high condensation). Thus, it is unlikely that condensation levels have an impact in EdU incorporation or detection."

This information is worth to be included in the discussion of the manuscript. Although I agreed with the authors that chromosome condensation might not be an important factor here to bias the results, it still worth to perform the control experiment to avoid any unexpected technical/biological reasons, which might bias the observation/detection. In addition, the control experiment is not difficult to perform.

2. In the response, the author indicated that "We have replaced "DNA replication" with "DNA synthesis" throughout the manuscript, except in certain contexts where we used "replication" in the context of our interpretation of the data."

Since the mechanism(s) of the observed DNA synthesis in late mitosis is not clear yet, it might be better not to use the term of "DNA replication" in the title, to avoid the confusion. The readers (in particular those not in the field of DNA replication) might think that the normal DNA replication

indeed happen after chromosome segregation.

For the same reason, I suggest to not use the term of replication in the following sentence: “

- in the abstract “yeast cells temporally overlap replication and chromosome segregation during normal growth”
- in page 3, “our data suggest that anaphase replication of specific genomic regions”
- in page 5, “showing that inhibition of replication in mitosis allows division of most”
- in page 9, “mitotic replication of diverse genomic regions, or mitotic replication of specific genomic regions”
- in page 13, “our data indicates that a substantial fraction of wild-type unstressed cells finish replication of specific chromosome regions late in mitosis”
- in page 16, “continued replication during late mitosis could contribute to the high frequency of ultrafine chromatin bridges” ... “Further, mitotic DNA replication may help explain how animal cells with rapid divisions such as in the early embryo”

3. I'm a bit confuse. Do the authors think the mitotic DNA synthesis that they observed in yeast depends on BIR (i.e. Pol32) or normal DNA polymerases, or both processes involve?

In discussion, the authors mentioned that “The idea that yeast cells complete DNA replication in anaphase has interesting parallels with mitotic DNA synthesis (MiDAS) observed in mammalian cells exposed to replication stress 6,39. Like yeast anaphase replication, MiDAS depends on the Pol32 homologue PolD3 and does not require Rad51, opening the possibility that a form of BIR plays a role in both processes.” Also, the author showed in Fig 5 the role of Pol32 in resolution of DNA bridges during anaphase. While in page 5, to try to address whether the EdU incorporation depends on DNA synthesis, the authors focus on the depletion of Pol3, which inhibited nuclear division and prevented formally demonstration. Why the authors do not try to use a Pol32 depletion, which shows less severe defects?

4. In Fig. S12a, it shows clearly that the link between under-replication and replication timing is very different between the subtelomeric regions and non-subtelomeric regions. The under-replication is only slightly increase (although significant) with replication timing in non-subtelomeric regions. In addition, we can observe a clear binominal distribution for the late-replicating regions (RT >50, non-subtelomeric regions), which suggests the existence of two distinguish populations, i.e late replicating under-replication and late replicating without under-replication. It will be interesting to further check what corresponds to these two populations. This result should move into the main text. Or not, based on the Fig 6d, the readers might have impression that there is a strong link between under-replication and replication timing, which indeed dominates by the under-replication of late-replicating subtelomeric regions.

5. The results of Fig S12b is important and should also be included into the main text.

And for the figures on the right (within subtelomeric regions), did the authors check the distribution of (i) their distances to the telomere and (ii) replication timing for different groups? To get a better idea what come from these differences.

6. As mentioned in previous comments, I have impression that the observed under-replication is indeed dominate by subtelomeric regions, although it has also been observed in few other late-replicating regions. This enrichment of under-replication at subtelomeric regions should be indicated in the abstract.

7. In response to the reviewer, the authors mentioned that “We tried BrdU-seq as suggested, although we anticipated technical difficulties with this approach, since our previous results predicted that only low levels of DNA synthesis occur during late mitosis. Indeed, although we did identify an enrichment of subtelomeric and rDNA sequences in some BrdUseq samples (consistent with our NGS data) we failed to obtain robust results in all replicates, perhaps due to the

extremely low amounts of DNA recovered in IPs from G1-arrested cells. We have therefore decided not to include these results, and cannot rule out that technical bias may be influencing the DNA sequencing data. On the other hand, since both EdU incorporation and time-lapse microscopy data support our model, and since the numbers from HU and pol2 inhibition (Fig 2) roughly match 40% from microscopy, we think that the interpretation that DNA synthesis during late mitosis is important for chromosome segregation is probably correct."

Please add it into the discussion of the manuscript.

8. As indicated by the authors "There is correlation between replication timing and mutation rates in yeast and human cells 24,25 and a strong relationship between distance to the telomere and mutation rate in budding yeast 26." in the manuscript, and "However, the mechanism behind higher mutation rates in these regions remained unclear. Our results suggested that late replication and/or segregation could be causing higher mutations in these regions." in the response to the reviewers.

I'm still not convince that the results shown by the authors bring new insights to the subject. The major point needs to be addressed here is that whether the higher mutation rates is due to distance to the telomere or due to under replication (or replication timing). And I don't think any of the current figures/results of the paper help to answer this question. For example, in Fig. 7b, instead of drawing the mean pattern for all under-replication regions, it will be more informatic to perform the analysis with different subgroups, i.e. the subtelomeric regions and non-subtelomeric regions, and maybe additional classes, transposable elements, fragile sites etc. (if it will have enough statistics). Since as observed by the authors, there are some under-replicating regions outside of subtelomeric regions, it will be important to check whether they show same higher mutation rates as those within subtelomeric regions, to help to confirm whether the under-replication (and not only the distance to the telomere) indeed contributes to the observed higher mutation rates. If the analysis confirms the role of under-replication in mutation rates, it can be further address whether the higher mutation rates will be specific for the under-replication regions but not due to the late replication per se, by comparison of the mutation rates between the under-replication regions (outside of subtelomeric regions) with the late-replicating regions of the same replication timing (e.g >50) but without under-replication (see point 4 for the description of these two groups).

And for the Fig 7d, it's a new result confirming that the role of distance to the telomere and/or replication timing is important on the mutation rates. It's not clear for me what are the replication timing for these four insertion points. We can image that the 2 kb one should be late, the 15 kb might be mid S phase, and the 100 kb and 150 kb might be early. Am I correct? Please provide this information in the manuscript.

However, again, this result did not confirm the role of under replication is important on the mutation rates. To really address the question, the authors need to insert the URA3 reporter into a telomeric distal under replication regions to check the mutation rates.

9. Minor point: in Fig 1b, using arrows of different colors (instead of arrows and arrowheads) might be better.

Reviewer #3 (Remarks to the Author):

The authors have thoughtfully addressed my comments.

We again thank the reviewers for their helpful comments. Find our responses below. Reviewer comments are in **bold**, and our replies in blue.

Reviewer #1

The revised manuscript by Ivanova and colleagues provides additional experimental data and explanations for all the issues I raised in the previous version. I particularly like the the alternative hypothesis(elevated dNTP levels during the previous S phase) proposed to explain the apparent paradox of the rad9- mutant.

I am very happy to endorse publication of this excellent manuscript. It is a fascinating analysis about how budding yeast cells complete DNA replication at an unexpected time in the cell cycle.

One minor comment in Figure S3 Legend: ‘Cells were growing in mid-log phase were treated...’

Thank you. We corrected the Figure S3 legend.

Reviewer #2

I’m glad that, in this revised version, the authors addressed most of the comments raised by the reviewers and improved significantly their manuscript by including additional results and figures, and the authors also made more attention on potential caveats in the interpretation of the results. I believe that this study will provide interesting results for the community, while there are still several important issues need to be addressed. These points are outlined below:

1. In the response to the author:

“In figure 1a, the authors observed that G1 cells showed higher amount of EdU incorporation than metaphase cells. I wonder whether the different condensation levels of chromosome between G1 and metaphase cells might have some impact on the EdU incorporation/detection. Can the authors provide some control showing that this observation is not due to the problem of condensation? For example, the authors can label the newly replicated DNA by EdU within S phase then check the EdU incorporation signals within the cells at S, G2, metaphase, anaphase and next G1 phase. Without any technical bias, the authors should be able to observe similar signals within cells at different phases. It will be better to compare directly the EdU incorporation level for the late mitotic cells (anaphase or telophase cells) and the metaphase cells as well.

We thank the reviewer for the suggestion. However, we think the proposed control is not necessary because there is no correlation between EdU levels and chromosome condensation. In budding yeast, chromosome condensation is low in G1, intermediate in metaphase, and reaches its maximum in anaphase (see for example Neurohr et al., Science 2011). In contrast, nuclear EdU incorporation is highest in G1 and late anaphase (when condensation is at its lowest and highest, respectively), and lowest in metaphase

and early anaphase (medium and high condensation). Thus, it is unlikely that condensation levels have an impact in EdU incorporation or detection.” This information is worth to be included in the discussion of the manuscript.

Although I agreed with the authors that chromosome condensation might not be an important factor here to bias the results, it still worth to perform the control experiment to avoid any unexpected technical/biological reasons, which might bias the observation/detection. In addition, the control experiment is not difficult to perform.

We performed the requested experiment (new figure **S2**). We synchronized cells in G1, added EdU for 10 minutes while the cells progressed through S, and then measured EdU incorporation during the subsequent cell-cycle (+15min, +30min, +45min), and visually staged cells. Other than the 2-fold drop in EdU as expected from nuclear division, we saw no strong differences in EdU signal. We also mention that chromosome condensation in yeast is highest in late anaphase:

Detection of EdU was not affected by differences in cell cycle stage such as chromosome condensation (which in budding yeast, is highest in late anaphase (Neurohr et al. 2011)), since EdU incorporated during S phase was detected with similar efficiency in mitosis and interphase (Fig. S2).

2. In the response, the author indicated that “We have replaced “DNA replication” with “DNA synthesis” throughout the manuscript, except in certain contexts where we used “replication” in the context of our interpretation of the data.” Since the mechanism(s) of the observed DNA synthesis in late mitosis is not clear yet, it might be better not to use the term of “DNA replication” in the title, to avoid the confusion. The readers (in particular those not in the field of DNA replication) might think that the normal DNA replication indeed happen after chromosome segregation. For the same reason, I suggest to not use the term of replication in the following sentence: “

- in the abstract “yeast cells temporally overlap replication and chromosome segregation during normal growth”
- in page 3, “our data suggest that anaphase replication of specific genomic regions”
- in page 5, “showing that inhibition of replication in mitosis allows division of most”
- in page 9, “mitotic replication of diverse genomic regions, or mitotic replication of specific genomic regions”
- in page 13, “our data indicates that a substantial fraction of wild-type unstressed cells finish replication of specific chromosome regions late in mitosis”
- in page 16, “continued replication during late mitosis could contribute to the high frequency of ultrafine chromatin bridges” ... “Further, mitotic DNA replication may help explain how animal cells with rapid divisions such as in the early embryo”

We changed the title to “Budding yeast complete DNA synthesis after chromosome segregation begins” and made the replacements requested:

- Abstract: Replaced with “Thus, yeast cells temporally overlap DNA synthesis and chromosome segregation during normal growth”

- P. 3, Replaced with “our data suggest that anaphase DNA synthesis of specific genomic regions”.
- P. 5, Replaced with “inhibition of DNA synthesis in mitosis”
- P. 9, 13, 16, Replaced “replication” with “DNA synthesis”

3. I’m a bit confuse. Do the authors think the mitotic DNA synthesis that they observed in yeast depends on BIR (i.e. Pol32) or normal DNA polymerases, or both processes involve?

In discussion, the authors mentioned that “The idea that yeast cells complete DNA replication in anaphase has interesting parallels with mitotic DNA synthesis (MiDAS) observed in mammalian cells exposed to replication stress 6,39. Like yeast anaphase replication, MiDAS depends on the Pol32 homologue PoID3 and does not require Rad51, opening the possibility that a form of BIR plays a role in both processes.” Also, the author showed in Fig 5 the role of Pol32 in resolution of DNA bridges during anaphase. While in page 5, to try to address whether the EdU incorporation depends on DNA synthesis, the authors focus on the depletion of Pol3, which inhibited nuclear division and prevented formally demonstration. Why the authors do not try to use a Pol32 depletion, which shows less severe defects?

As shown in figure 5c, resolution of most anaphase bridges requires “normal” DNA synthesis (Pol2 and Pol3) whereas Pol32 is required for resolution of a smaller fraction of bridges. This suggests that “Pol32-independent bridge resolution mechanisms must exist”, as stated in the discussion. To clarify, we have changed this to “Pol32-independent bridge resolution mechanisms must *also* exist.”

Concerning the second point: we did not try to use Pol32 mutants in the EdU experiments and chose Pol3 instead because we wanted to completely inhibit DNA synthesis. Depletion of Pol32 would have at most a partial effect, since it is not an essential protein, and negative results with these mutants could have been inconclusive.

4. In Fig. S12a, it shows clearly that the link between under-replication and replication timing is very different between the subtelomeric regions and non-subtelomeric regions. The under-replication is only slightly increase (although significant) with replication timing in non-subtelomeric regions. In addition, we can observe a clear binominal distribution for the late-replicating regions (RT >50, non-subtelomeric regions), which suggests the existence of two distinguish populations, i.e late replicating under-replication and late replicating without under-replication. It will be interesting to further check what corresponds to these two populations. This result should move into the main text. Or not, based on the Fig 6d, the readers might have impression that there is a strong link between under-replication and replication timing, which indeed dominates by the under-replication of late-replicating subtelomeric regions.

We agree. We have re-analyzed these data to better understand the relationship between replication timing and under-replication. In the process of doing this, we realized that we had

inadvertently included rDNA genes in the graphs in figure 6d and S12a. We excluded these genes from all other analysis since differences in DNA copy number cannot be distinguished from differences in rDNA repeat copy number. Our new analysis is presented in a new figure 6d and S12. Note that we moved some data from the old figure S12a into the main figure, and distinguished data between subtelomeric and non-subtelomeric regions. (To simplify the new figure S12, we removed data from multiple sources we had included in previous version and concentrated in showing data from one single source, that we judged most robust). This shows that all loci with rep-timing > 50 min show significantly higher under-replication. The highest level of under-replication is found in subtelomeric regions that have late replication timing. Late-replicating non-subtelomeric regions are restricted to a region in chromosome IVR (from 981 to 989 kb), which is also under-replicated. Interestingly, this region contains a cluster of transposable elements. Therefore, under-replication and late-replication are tightly correlated, both near chromosome ends and in chromosome arms. The main text now reads:

All loci with a measured replication timing greater than 50 minutes, both subtelomeric and non-subtelomeric, show significantly higher under-replication (Fig. 6d).

5. The results of Fig S12b is important and should also be included into the main text. And for the figures on the right (within subtelomeric regions), did the authors check the distribution of (i) their distances to the telomere and (ii) replication timing for different groups? To get a better idea what come from these differences.

We present the results from former Figure S12b, as well as new analysis along the lines suggested by the reviewer in new panels e-f in figure 6. This shows clearly that transposable elements and fragile sites are mostly under-replicated and located away from telomeres, whereas G4-rich regions are mostly under-replicated and near telomeres. Overall, we think that panels d, e and f in the new figure 6 clarify that under-replication occurs mostly on G4-rich regions in subtelomeric regions, but that under-replication is also detected away from telomeres, in regions containing transposable elements and fragile sites.

6. As mentioned in previous comments, I have impression that the observed under-replication is indeed dominate by subtelomeric regions, although it has also been observed in few other late-replicating regions. This enrichment of under-replication at subtelomeric regions should be indicated in the abstract.

We agree and have changed the abstract to read:

...the decrease in cyclin-Cdk activity during mitotic exit allows DNA synthesis to finish at subtelomeric and difficult-to-replicate regions.

7. In response to the reviewer, the authors mentioned that “We tried BrdU-seq as suggested, although we anticipated technical difficulties with this approach, since our previous results predicted that only low levels of DNA synthesis occur during late mitosis. Indeed, although we did identify an enrichment of subtelomeric and rDNA sequences in some BrdUseq samples (consistent with our NGS data) we failed to obtain

robust results in all replicates, perhaps due to the extremely low amounts of DNA recovered in IPs from G1-arrested cells. We have therefore decided not to include these results, and cannot rule out that technical bias may be influencing the DNA sequencing data. On the other hand, since both EdU incorporation and time-lapse microscopy data support our model, and since the numbers from HU and pol2 inhibition (Fig 2) roughly match 40% from microscopy, we think that the interpretation that DNA synthesis during late mitosis is important for chromosome segregation is probably correct.”Please add it into the discussion of the manuscript.

We have included the following in the discussion:

Together, our data indicates that a substantial fraction of wild-type unstressed cells complete DNA synthesis late in mitosis, long after the initiation of chromosome segregation. In particular, EdU incorporation and time-lapse imaging experiments indicate that DNA synthesis stops or slows down during metaphase and resumes during late anaphase or early G1. In addition, DNA copy number analysis suggests that difficult-to-replicate sequences are frequently under-replicated in metaphase, especially G-quadruplexes near telomeres, and a subset of transposable elements and fragile sites in non-subtelomeric regions. It will be interesting to confirm this observation with independent methods such as sequencing of DNA synthesized in anaphase / G1.

8. As indicated by the authors “There is correlation between replication timing and mutation rates in yeast and human cells 24,25 and a strong relationship between distance to the telomere and mutation rate in budding yeast 26.” in the manuscript, and “However, the mechanism behind higher mutation rates in these regions remained unclear. Our results suggested that late replication and/or segregation could be causing higher mutations in these regions.” in the response to the reviewers.

I’m still not convinced that the results shown by the authors bring new insights to the subject. The major point needs to be addressed here is that whether the higher mutation rates is due to distance to the telomere or due to under replication (or replication timing). And I don’t think any of the current figures/results of the paper help to answer this question. For example, in Fig. 7b, instead of drawing the mean pattern for all under-replication regions, it will be more informative to perform the analysis with different subgroups, i.e. the subtelomeric regions and non-subtelomeric regions, and maybe additional classes, transposable elements, fragile sites etc. (if it will have enough statistics). Since as observed by the authors, there are some under-replicating regions outside of subtelomeric regions, it will be important to check whether they show same higher mutation rates as those within subtelomeric regions, to help to confirm whether the under-replication (and not only the distance to the telomere) indeed contributes to the observed higher mutation rates. If the analysis confirms the role of under-replication in mutation rates, it can be further address whether the higher mutation rates will be specific for the under-replication regions but not due to the late replication per se, by

comparison of the mutation rates between the under-replication regions (outside of subtelomeric regions) with the late-replicating regions of the same replication timing (e.g >50) but without under-replication (see point 4 for the description of these two groups).

And for the Fig 7d, it's a new result confirming that the role of distance to the telomere and/or replication timing is important on the mutation rates. It's not clear for me what are the replication timing for these four insertion points. We can image that the 2 kb one should be late, the 15 kb might be mid S phase, and the 100 kb and 150 kb might be early. Am I correct? Please provide this information in the manuscript.

However, again, this result did not confirm the role of under replication is important on the mutation rates. To really address the question, the authors need to insert the URA3 reporter into a telomeric distal under replication regions to check the mutation rates.

The reviewer makes important points about the relationship between replication timing, distance to the telomere, and mutation rates, while stating that our findings do not bring new insights. We respectfully disagree, but understand the confusion.

Our findings indicate that mutation rates (in the lab and the wild) are correlated with both distance to telomeres and replication after metaphase. However, because relatively few non-subtelomeric loci are under-replicated in metaphase, the data do not allow to differentiate among distance-from-the-telomere vs replication-timing vs under-replication as being the largest cause of increased mutation rates (see also discussion of our new analysis at the end of this section). New fluctuation assays interrogating other chromosome loci proposed by the reviewer could help clarify whether distance to the telomere or replication after metaphase cause high mutation rates. However, we did not intend to establish causality and consider these experiments out of the scope of our paper. We have gone over the text and made sure that we avoid implying that we know what is causal for high mutation rates in the subtelomeres.

Instead, the new insight provided by our findings is the identification of two possible causes for high mutation rates in subtelomeric regions: error-prone replication after metaphase, and damage of DNA bridges by cytokinesis. The first one depends on replication timing, the second could depend on either replication timing or proximity to telomeres (because telomeres segregate last, whether replicated or not; in addition, under-replication could contribute to their late segregation). We have made these points more explicit in the main text.

We also must emphasize that to our knowledge, the decrease in mutation rate in subtelomeric regions in the *cyk3* background is the first data showing that a delay in cytokinesis causes a lower mutation rate (and may help explain the results in Vinton and Weinert 2017). This is consistent with a need to finish DNA synthesis during late mitosis.

All this being said, we have performed additional analysis to further clarify specific issues raised by the reviewer:

- Figure 7b (previous version) showed the relationship between SNPs and InDels and under-replication. As suggested, we have performed the same analysis in different regions (split by replication timing or by distance to the telomere). The higher fraction of SNPs and InDels in under-replicated regions is mostly driven by the proximity to telomeres and/or replication timing. We have moved these results to a new Figure S15. We have updated the manuscript accordingly to highlight this issue.

- For figure 7d, we have added information on the replication timing and under-replication of the loci analysed.

- We also added additional analysis of the relationship between the fitness cost of deleting each gene, and proximity to telomeres and replication timing (Figure S14). Genes that exhibit a high fitness cost when knocked out show lower under-replication and are positioned further from telomeres but do not show different replication timing (when compared to genes that are ~dispensable i.e. show low cost once knocked out). We have updated the manuscript accordingly.

9. Minor point: in Fig 1b, using arrows of different colors (instead of arrows and arrowheads) might be better.

We have done so.

Reviewer #3:

The authors have thoughtfully addressed my comments.

Reviewers' comments:

Reviewer #2 (Remarks to the Author):

In this new revised version, the authors addressed almost all the points that I raised. They include significant new analysis results and figures. Also, some sections of the manuscript (i.e. the results and discussion associated with the analyses on the mutation frequency) were significantly rewritten, which makes the discussion easy to follow. I still believe that this is an interesting work, while some results obtained by the analyses that I requested do not support some of their key conclusions. These important issues have to be carefully and clearly addressed before publication. These points are outlined below:

1. In the Fig. 6d, it clearly shows that all subtelomeric regions present higher under-replication than non-subtelomeric regions, for all replication timing groups. Even for the subtelomeric regions with a timing <20 min, their under-replication levels are much higher than almost all non-subtelomeric regions, including those have a much later replication timing, e.g. 30-40 min. Moreover, the under-replication levels are similar for all replication timing groups for the non-subtelomeric regions (except one region with timing >50 min, see below). These results indicate that the observed dependence of under-replication on replication timing is a specific feature for the subtelomeric regions, and is not a general feature.

2. Based on the Supplementary Fig. 10 and the observations on Fig. 6d discussed in point 1, it seems that, instead of replication timing, it's the distance to the telomere is important for the under-replication level. I wonder whether there is a hidden correlation between replication timing and distance to telomere, which can explain the observed dependence of under-replication on replication timing for the subtelomeric regions. The heatmap scale indicating the distance to the closest telomere on Fig. 6d do not have enough resolution to answer this question. Please further check and verify this important issue.

3. In the response to the author:

"Late-replicating non-subtelomeric regions are restricted to a region in chromosome IVR (from 981 to 989 kb), which is also under-replicated. Interestingly, this region contains a cluster of transposable elements. Therefore, under-replication and late-replication are tightly correlated, both near chromosome ends and in chromosome arms."

Since it's only one specific region containing a cluster of transposable elements (Which TE? Would you please specify?) show extremely late replication timing (i.e. >50 min) and strong under replication, while the analysis on all other non-subtelomeric regions does not support a link between under-replication and replication timing (see point 1). We do observe that the non-subtelomeric regions showing the highest under-replication level (e.g >20%) enrich at the groups with mid-replication timing, i.e. 20-30 and 30-40 min. Based on all these observations, I don't think the authors can draw such strong conclusion and claim that "under-replication and late-replication are tightly correlated".

Please add, in the manuscript, the description of this region with timing >50 min. It's important for the readers know that all points of this group come from one specific genomic region with some particular feature. The under-replication of this region might not relate to the late replication timing at all.

4. Additionally, on the response to the author:

"We have re-analyzed these data to better understand the relationship between replication timing and under-replication. In the process of doing this, we realized that we had inadvertently included

rDNA genes in the graphs in figure 6d and S12a. We excluded these genes from all other analysis since differences in DNA copy number cannot be distinguished from differences in rDNA repeat copy number."

As described by the authors, other late-replicating regions with timing >50 min belong to genomic loci containing rDNA genes. And, if I remember correctly, the results in previous manuscript shown that these regions are not under-replicated. - I suppose that these regions correspond to the points on the bottom, i.e. with low under replication level, of the bimodal distribution observed for the late replicating group in the Fig. S11 of previous manuscript; am I correct?

I understand that we cannot distinguish between different rDNA repeats. But we should be able to observe the average behavior of rDNA repeats. Since, the analysis results indicate that, in average, rDNA regions are late-replicating, I don't understand why we could not detect the under-replication within these regions if it was the case. Can the authors explain the reason behind that? I think, instead of remove this result from the manuscript, it is important to mention this result in the discussion and maybe with a supplementary figure. Again, this result shows that all late-replicating regions are not under replicated, and it does not exist a direct link between under-replication and replication timing.

5. The main text now reads: "All loci with a measured replication timing greater than 50 minutes, both subtelomeric and non-subtelomeric, show significantly higher under-replication (Fig. 6d)."

Based on all the points I raised in 1-4, I doubt whether it still makes sense to conclude that "All loci with a measured replication timing greater than 50 minutes, both subtelomeric and non-subtelomeric, show significantly higher under-replication (Fig. 6d)." This will create the confusion and make the readers think that the replication timing is the key factor, which is not really supported by the data.

6. Also, for the same reason, please remove the "late replicating" in the section title on page 9, "Late replicating, sub-telomeric, and difficult-to-replicate sequences are replicated".

7. Based on the new results shown in Fig. 6d and Fig. S15, it strongly indicates that the mutation rates (both SNPs or InDels) are not depend on the under-replication level nor on the replication timing, for the non-subtelomeric regions. Several observations support this conclusion:

(i.) The group with the lowest under-replication level (i.e. <15) display highest SNP and InDel frequencies (Fig. S15 Top).

(ii.) The distribution of SNP and InDel frequencies are similar for each replication timing group.

(iii.) The only locus with timing >50 min showing high under replication level (Fig. 6d), has very low SNP and InDel frequencies (Fig. 15 Bottom).

8. On the other hand, it is weird that the authors did not observe a dependence of SNP density on replication timing. This correlation is well-established by many studies in many organisms including yeast (see for example, Agier, N. & Fischer, G. Mol. Biol. Evol. 2012). It might due to the fact that the authors used the SNPs in all genomic positions in their analysis instead of using those within synonymous and intergenic sites as others did in previous studies. Since the non-synonymous sites within genic regions have strong functional constraints, they need to be analysed separately. Or not, it might lead to wrong conclusion. Such separation is even more critical for the analysis in yeast with a very compact genome.

9. For the subtelomeric regions, again, the increase of SNP and InDel frequencies (Fig. S15) in

function of under replication level (and replication timing) could be explained by the distance to telomere. As mentioned by the author, the distance to telomere is a well-established parameter strongly impacts diversity due to the bias gene conversion, with regions closer to telomere (thus later replication and higher under) show higher substitution rates. Although the authors claim that they don't want to draw a conclusion on causality, they intend to draw the conclusion that under-replication indeed contributes in increasing mutation rates and might play a positive role in evolution. However, this conclusion is not supported by their results.

10. In the abstract, "DNA synthesis during late mitosis correlates with elevated mutation rates, including copy number variation. Thus..."

I don't think the authors can make such strong statement in their abstract. As discussed in points 7-9, this conclusion is not supported by their data. Please revise it.

For the same reason, please revise the following sentence on page 3 "suggest that anaphase DNA synthesis of specific genomic regions, notably near chromosomes ends, may contribute to their high mutation rates and rapid evolutionary diversity."

11. There are still some places using "DNA replication" instead of "DNA synthesis", in particular, within the new texts added in the revised version. Again, I suggest to replace "DNA replication" by "DNA synthesis" in the following sentences: "

- in page 11, "Replication after metaphase and late anaphase segregation may contribute to high rates of evolutionary divergence in subtelomeric regions"

- in page 15, "The first one depends on their replication after metaphase..."

- in page 27, "DNA replication is inhibited by high M-Cdk levels during metaphase and resumes during late mitosis..."; and also, on the Fig. "Replication of late regions..."

Minor points:

12. Page 2, Abstract, "mitotic exit allows DNA synthesis to finish at subtelomeric and difficult-to-replicate regions"; rather use "...and some difficult-to-replicate regions".

13. Page 4, "for DNA synthesis 16,17 we chose to deplete the...", add a common "," before "we".

14. Page 12, "These results support the hypothesis that the increased mutation rate of late-replicating, late-segregating chromosome regions is due to..." change into "...support the hypothesis that the increased frequency of URA3 loss at subtelomeric chromosome regions is due to...". I don't think the authors can generalize their result here.

15. Page 13, "replication and chromosome arm length are negatively correlated ($r=-0.65$) (Figure S16a)." When you check carefully the Fig. S 16a, this correlation is obviously overestimated. On the Fig. S16a, it clearly shows that the chromosome arms with extremely short length (i.e. < 100 kb) present the largest length of under replication. If you remove these points, the reported negative correlation is almost gone.

16. Page 24, Fig. 6a, in the figure legend, it indicates "late anaphase", but in the figure, it is "telophase". Should be better using the same term.

We thank the reviewer again for the thoughtful comments. Our replies below are in blue, the reviewer comments are in **bold**.

Reviewer #2 (Remarks to the Author):

In this new revised version, the authors addressed almost all the points that I raised. They include significant new analysis results and figures. Also, some sections of the manuscript (i.e. the results and discussion associated with the analyses on the mutation frequency) were significantly rewritten, which makes the discussion easy to follow. I still believe that this is an interesting work, while some results obtained by the analyses that I requested do not support some of their key conclusions. These important issues have to be carefully and clearly addressed before publication. These points are outlined below:

1. In the Fig. 6d, it clearly shows that all subtelomeric regions present higher under-replication than non-subtelomeric regions, for all replication timing groups. Even for the subtelomeric regions with a timing <20 min, their under-replication levels are much higher than almost all non-subtelomeric regions, including those have a much later replication timing, e.g. 30-40 min. Moreover, the under-replication levels are similar for all replication timing groups for the non-subtelomeric regions (except one region with timing >50 min, see below). These results indicate that the observed dependence of under-replication on replication timing is a specific feature for the subtelomeric regions, and is not a general feature.

We agree that the correlation between under-replication in mitosis and replication timing is not a general feature, and that it is almost entirely restricted to subtelomeric regions. We have rewritten the results section to clarify this, as shown below. (We have also included a description of the region with replication timing >50 min in chromosome IV, as requested in point 3).

Most regions under-represented in mitosis correspond to a subset of late-replicating regions, most, but not all, of which are subtelomeric (**Figure 6a-d, S12**). The relationship between late replication timing and under-representation in mitosis is restricted to subtelomeric regions, with the exception of one 10-kb region in the middle of chromosome IV (from 981 to 989 kb) containing two transposable elements and two tRNA genes, which has a very late replication timing (>50 minutes) (**Figure 6d**). Additionally, difficult-to-replicate regions such as transposable elements, fragile sites²⁴ and loci predicted to contain G-quadruplexes show significant under-representation in metaphase (**Figure 6e**). We conclude that most regions near telomeres, and a relatively smaller number of loci in non-subtelomeric regions, are under-represented in mitosis and refer to these regions as “under-replicated in mitosis”.

2. Based on the Supplementary Fig. 10 and the observations on Fig. 6d discussed in point 1, it seems that, instead of replication timing, it's the distance to the telomere is important for the under-replication level. I wonder whether there is a hidden correlation between replication timing and distance to telomere, which can explain the observed dependence of under-replication on replication timing for the subtelomeric regions. The heatmap scale indicating the distance to the closest telomere on Fig. 6d do not have enough resolution to answer this question. Please further check and verify this important issue.

Is there a hidden correlation between replication timing and distance to the telomere? This is an interesting question. In short, we have not found any. To check if replication timing is a significant feature even when taking into account distance from the telomere, and to check for any hidden correlations between distance from the telomere and %underrep that might be causal for the relation between replication timing and %underrep, we used two approaches. First, we used a linear regression model to determine if replication timing is a significant predictor of %underrep after taking into account the distance to the telomere. We find that for all thresholds for subtelomeric, replication timing is always a significant predictor, even after taking into account distance to the telomere (**A**). Other models such as random forest and non-linear regression models gave the same result (not shown).

Second, we used a moving median to remove the relation between %underrep and distance to the telomere, and asked if %underrep can explain the difference between the measured %underrep and the moving median value (this method is commonly used for noise in gene expression, eg: Newman et al. 2006, doi:10.1038/nature04785). The resulting 'DM' value (shown in **B**) is the %underrep after taking into account the distance to the telomere. (**C**) shows that %underrep-DM (black) is correlated with replication timing in subtelomeric regions. Red shows the correlation. These results are shown in a new Supplementary Figure 12b-c.

A $\% \text{under-replication}_{\text{CDC20}} = \% \text{GC} + \text{Trep} + \text{dist-to-tel} + \log(\text{dist-to-tel}) + \text{G4}$

In summary: while absence of evidence is not evidence of absence, we did not find any evidence for a hidden correlation between replication timing and distance to telomere which can explain the observed dependence of under-replication on replication timing for the subtelomeric regions.

3. In the response to the author: “Late-replicating non-subtelomeric regions are restricted to a region in chromosome IVR (from 981 to 989 kb), which is also under-replicated. Interestingly, this region contains a cluster of transposable elements. Therefore, under-replication and late-replication are tightly correlated, both near chromosome ends and in chromosome arms.”

Since it’s only one specific region containing a cluster of transposable elements (Which TE? Would you please specify?) show extremely late replication timing (i.e. >50 min) and strong under replication, while the analysis on all other non-subtelomeric regions does not support a link between under-replication and replication timing (see point 1). We do observe that the non-subtelomeric regions showing the highest under-replication level (e.g >20%) enrich at the groups with mid-replication timing, i.e. 20-30 and 30-40 min. Based on all these observations, I don’t think the authors can draw such strong conclusion and claim that “under-replication and late-replication are tightly correlated”.

Please add, in the manuscript, the description of this region with timing >50 min. It's important for the readers know that all points of this group come from one specific genomic region with some particular feature. The under-replication of this region might not relate to the late replication timing at all.

As mentioned in our reply to point 1, we agree with the reviewer's conclusions and have clarified this in the results section. We have included a description of the region with replication timing >50 min in chromosome IV.

4. Additionally, on the response to the author: "We have re-analyzed these data to better understand the relationship between replication timing and under-replication. In the process of doing this, we realized that we had inadvertently included rDNA genes in the graphs in figure 6d and S12a. We excluded these genes from all other analysis since differences in DNA copy number cannot be distinguished from differences in rDNA repeat copy number."

As described by the authors, other late-replicating regions with timing >50 min belong to genomic loci containing rDNA genes. And, if I remember correctly, the results in previous manuscript shown that these regions are not under-replicated. - I suppose that these regions correspond to the points on the bottom, i.e. with low under replication level, of the bimodal distribution observed for the late replicating group in the Fig. S11 of previous manuscript; am I correct?

I understand that we cannot distinguish between different rDNA repeats. But we should be able to observe the average behavior of rDNA repeats. Since, the analysis results indicate that, in average, rDNA regions are late-replicating, I don't understand why we could not detect the under-replication within these regions if it was the case. Can the authors explain the reason behind that? I think, instead of remove this result from the manuscript, it is important to mention this result in the discussion and maybe with a supplementary figure. Again, this result shows that all late-replicating regions are not under replicated, and it does not exist a direct link between under-replication and replication timing.

The reviewer is correct that the original version of Figure S11 assigned low under-replication to rDNA sequences, which are late-replicating. We remain reluctant to include rDNA copy number data in the manuscript, for reasons explained in detail below. Ultimately however, the important issue here is whether these data support a general relationship between under-replication and replication timing. We agree with the reviewer that there is no general relationship between these two variables outside of the subtelomeric regions (see our reply to point 1); including potentially misleading rDNA data to bolster a point we are already making elsewhere (with better data) would be confusing.

Regarding rDNA copy number data: multiple studies show that neither absolute nor relative quantification of rDNA copy number by Illumina sequencing gives reliable results. For instance, "WGS ... produce high error in copy number estimation even for technical replicates" (Morton et al., Challenges and Approaches to Genotyping Repetitive DNA. G3. 2020). Below is a modified figure from Morton et al. showing the effect of DNA input into library prep. They observe both high errors rates (large variance in the Y axis at each point on the X axis), and systematic biases due to input DNA and actual rDNA copy number.

Figure 1D reproduced from Morton et al., Challenges and Approaches to Genotyping Repetitive DNA, G3: GENES, GENOMES, GENETICS January 1, 2020 vol. 10 no. 1 417-430; <https://doi.org/10.1534/g3.119.400771> CC BY 4.0

Lofgren et al. (Genome-based estimates of fungal rDNA copy number variation across phylogenetic scales and ecological lifestyles. Molecular Ecology. 2019) found similar levels of very high variance in rDNA copy number across replicates, as measured by Illumina whole genome sequencing (their Table S2). Their measurements of rDNA copy number vary by up to 70% between sequencing lanes. The reasons for this are not understood. An additional factor is that rDNA copy number varies even within cells in one clonal population (visible as smear bands in CHEF gels, see for example Kobayashi and Sasaki 2017, doi.org/10.1093). Finally, standard WGS cannot differentiate rDNA circles from genomic rDNA.

That being said, the figure below shows the results of our analysis of rDNA copy number for three independent experiments.

CDK inhibition in Cdc20-depleted cells increases rDNA copy number, as would be expected if rDNA is under-replicated in M phase and CDK inhibition allows rDNA synthesis ($p=0.03$, t-test). However against this interpretation, we see no significant difference in rDNA copy number between metaphase and G1 samples in two independent experiments ($p>0.05$), suggesting that rDNA is not under-replicated in M phase. Thus, the results are inconclusive. Given the above mentioned caveats associated with rDNA copy number analysis, we prefer not to include these data in the paper.

5. The main text now reads: “All loci with a measured replication timing greater than 50 minutes, both subtelomeric and non-subtelomeric, show significantly higher under-replication (Fig. 6d).”

Based on all the points I raised in 1-4, I doubt whether it still makes sense to conclude that “All loci with a measured replication timing greater than 50 minutes, both subtelomeric and non-subtelomeric, show significantly higher under-replication (Fig. 6d).” This will create the confusion and make the readers think that the replication timing is the key factor, which is not really supported by the data.

We have removed this statement, and the relevant paragraph was modified as described in our reply to point 1.

6. Also, for the same reason, please remove the “late replicating” in the section title on page 9, “Late replicating, sub-telomeric, and difficult-to-replicate sequences are replicated”.

We have modified the text as requested.

7. Based on the new results shown in Fig. 6d and Fig. S15, it strongly indicates that the mutation rates (both SNPs or InDels) are not depend on the under-replication level nor on the replication timing, for the non-subtelomeric regions. Several observations support this conclusion:

(i.) The group with the lowest under-replication level (i.e. <15) display highest SNP and InDel frequencies (Fig. S15 Top).

(ii.) The distribution of SNP and InDel frequencies are similar for each replication timing group.

(iii.) The only locus with timing >50 min showing high under replication level (Fig. 6d), has very low SNP and InDel frequencies (Fig. 15 Bottom).

We agree; and see our reply to the next related question.

8. On the other hand, it is weird that the authors did not observe a dependence of SNP density on replication timing. This correlation is well-established by many studies in many organisms including yeast (see for example, Agier, N. & Fischer, G. Mol. Biol. Evol. 2012). It might be due to the fact that the authors used the SNPs in all genomic positions in their analysis instead of using those within synonymous and intergenic sites as others did in previous studies. Since the non-synonymous sites within genic regions have strong functional constraints, they need to be analysed separately. Or not, it might lead to wrong conclusion. Such separation is even more critical for the analysis in yeast with a very compact genome.

We also thought it was odd that we did not observe a dependence of SNP density on replication timing. We therefore reanalyzed the data from Agier & Fischer 2012. We first reproduced their Fig1A showing a strong correlation with replication timing and SNP density (A,C). We then repeated the same analysis after splitting the SNPs into two groups: subtelomeric (blue) and non-subtelomeric (red) (B,D). This split largely removes the correlation for non-subtelomeric regions; sub-telomeric regions have higher SNP densities. See our new supplementary figure 15, with Fig1A from Agier reproduced below. The differences in numeric values between their figure and ours because they normalized so that the sum of all bins is 100%; we do not because this makes a quantitative comparison between subtelomeric and non-subtelomeric impossible :

Figure 1A reproduced from Agier & Fischer et al., The Mutational Profile of the Yeast Genome Is Shaped by Replication, *Molecular Biology and Evolution*, Volume 29, Issue 3, March 2012, Pages 905–913, <https://doi.org/10.1093/molbev/msr280> by permission of Oxford University Press

9. For the subtelomeric regions, again, the increase of SNP and InDel frequencies (Fig. S15) in function of under replication level (and replication timing) could be explained by the distance to telomere. As mentioned by the author, the distance to telomere is a well-established parameter strongly impacts diversity due to the bias gene conversion, with regions closer to telomere (thus later replication and higher under) show higher substitution rates. Although the authors claim that

they don't want to draw a conclusion on causality, they intend to draw the conclusion that under-replication indeed contributes in increasing mutation rates and might play a positive role in evolution. However, this conclusion is not supported by their results.

We show that under-replication is associated with subtelomeric regions, which have increased mutation rates; and that delaying cytokinesis reduces mutation rates in subtelomeric regions. Based on these data, we suggest that cytokinesis-dependent damage of subtelomeres may be a cause of mutations. To clarify this, we have rephrased the last sentence in the results section:

Interestingly, the *cyk3Δ* mutant displayed reduced *URA3* loss specifically in telomere-proximal loci, but not in telomere-distal positions (**Figure 7c**). These results support the hypothesis that the increased frequency of *URA3* loss at subtelomeric chromosome regions is due, at least in part, to cytokinesis-dependent DNA damage.

10. In the abstract, “DNA synthesis during late mitosis correlates with elevated mutation rates, including copy number variation. Thus...”

I don't think the authors can make such strong statement in their abstract. As discussed in points 7-9, this conclusion is not supported by their data. Please revise it.

For the same reason, please revise the following sentence on page 3 “suggest that anaphase DNA synthesis of specific genomic regions, notably near chromosomes ends, may contribute to their high mutation rates and rapid evolutionary diversity.”

We have modified the abstract and introduction to specify that this correlation is restricted to subtelomeric regions:

[Abstract] DNA synthesis during late mitosis correlates with elevated mutation rates at subtelomeric regions, including copy number variation.

[introduction] (...) our data suggest that anaphase DNA synthesis of chromosome end regions may contribute to their high mutation rates and rapid evolutionary diversity.

11. There are still some places using “DNA replication” instead of “DNA synthesis”, in particular, within the new texts added in the revised version. Again, I suggest to replace “DNA replication” by “DNA synthesis” in the following sentences: “

- in page 11, “Replication after metaphase and late anaphase segregation may contribute to high rates of evolutionary divergence in subtelomeric regions”

- in page 15, “The first one depends on their replication after metaphase...”
- in page 27, “DNA replication is inhibited by high M-Cdk levels during metaphase and resumes during late mitosis...”; and also, on the Fig. “Replication of late regions...”

We have done the changes as requested.

Minor points:

12. Page 2, Abstract, “mitotic exit allows DNA synthesis to finish at subtelomeric and difficult-to-replicate regions”; rather use “...and some difficult-to-replicate regions”.

We have changed the text as requested.

13. Page 4, “for DNA synthesis 16,17 we chose to deplete the...”, add a common “,” before “we”.

We have changed the text as requested.

14. Page 12, “These results support the hypothesis that the increased mutation rate of late-replicating, late-segregating chromosome regions is due to...” change into “...support the hypothesis that the increased frequency of URA3 loss at subtelomeric chromosome regions is due to...”. I don’t think the authors can generalize their result here.

We have changed the text as requested.

15. Page 13, “replication and chromosome arm length are negatively correlated ($r=-0.65$) (Figure S16a).” When you check carefully the Fig. S 16a, this correlation is obviously overestimated. On the Fig. S16a, it clearly shows that the chromosome arms with extremely short length (i.e. < 100 kb) present the largest length of under replication. If you remove these points, the reported negative correlation is almost gone.

Taking only arms of length > 100kb, as suggested, does not significantly change the correlation (from -0.6471 to -0.6244).

Being even more restrictive, and taking a very limited central subset (boxed region) does slightly reduce the correlation, but the correlation remains significant. We used bootstrapping (random sampling with replacement) to determine if outliers drive the high correlation. If the correlation is driven by only a few outlier datapoints, then many bootstraps will have a correlation of zero.

Show below are histograms of correlations from bootstrapping the data for all arms (left) and for the central subset (right). The results of these analyses have been added to the legend of figure S16.

16. Page 24, Fig. 6a, in the figure legend, it indicates “late anaphase”, but in the figure, it is “telophase”. Should be better using the same term.

We have changed the text as requested.

REVIEWERS' COMMENTS:

Reviewer #2 (Remarks to the Author):

The authors have addressed all my concerns, and the revised manuscript is suitable for publication.